# A first comparison of TROPOMI aerosol layer height to CALIOP data

Swadhin Nanda[2], Martin de Graaf[1], J. Pepijn Veefkind[1, 2], Maarten Sneep[1], Mark ter Linden[1, 3], Jiyunting Sun[1, 2], and Pieternel F. Levelt[1, 2]

[1]Royal Netherlands Meteorological Institute (KNMI), Utrechtseweg 297, 3731 GA De Bilt, The Netherlands
[2]Delft university of Technology (TU Delft), Mekelweg 2, 2628 CD Delft, The Netherlands
[3]S&T Corp, Delft, The Netherlands

*Correspondence to:* Swadhin Nanda (s.nanda@tudelft.nl)

**Abstract.** The Tropospheric Monitoring Instrument's (TROPOMI) level-2 aerosol layer height (ALH) product has now been released to the general public. This product is retrieved using TROPOMI's measurements of the oxygen A-band, radiative transfer model (RTM) calculations augmented by neural networks and an iterative optimal estimation technique. The TROPOMI ALH product will deliver aerosol layer height estimates over cloud-free scenes over the ocean and land that contain aerosols above a certain threshold of the measured UV absorbing index (UVAI) in the ultraviolet region. This paper provides background for the ALH product and explores its quality by comparing ALH estimates to similar quantities derived from spaceborne lidars observing the same scene. The spaceborne lidar chosen for this study is the Cloud-Aerosol Lidar with Orthogonal Polarisation (CALIOP) on board the Cloud-Aerosol Lidar and Infrared Pathfinder Satellite Observations (CALIPSO) mission, which flies in formation with NASA's A-train constellation since 2006 and is a proven source of data for studying aerosol layer heights. The influence of the surface and clouds are discussed and the aspects of the TROPOMI ALH algorithm that will require future development efforts are highlighted.

## 1 Introduction

Aerosols are an important component of the Earth system which provide the means for the formation of clouds by acting as cloud-condensation nuclei, affecting the Earth's radiation budget by absorbing or scattering incoming solar radiation (Twomey, 1974), and even nurturing forests from across oceans (Yu et al., 2015; Barkley et al., 2019). A significant source of origin for aerosols is natural, followed by anthropogenic contribution to the Earth's atmosphere. The IPCC (2014) report has made it clear that the current scientific consensus acknowledges the impact of aerosols on the Earth's radiation budget via direct, indirect and semi-direct effects. What makes monitoring aerosols difficult is the high spatial and temporal variability of aerosol micro and macrophysical properties (Li et al., 2009). To that extent, there are several spaceborne, ground-based and airborne missions extensively monitoring these aerosol micro and macrophysical properties. These missions aim to reduce the gaps in our knowledge of aerosol radiative effects by accurately measuring aerosol properties at a high spatial and temporal resolution. This paper specifically discusses retrieving information on the vertical distribution of aerosol layer in the atmosphere, which has significant relevance in deriving auxiliary aerosol properties and subsequently understanding aerosol radiative effects (ARE),

primarily absorption of radiation by aerosols. Torres et al. (1998) explicitly mention the importance of knowledge about aerosol vertical distribution which can be used in tandem with the UV absorbing index (UVAI) to compute aerosol properties such as aerosol optical thickness and effective single scattering albedo over cloud-free and snow-free scenes. de Graaf (2005) provide several sensitivity analyses that detail the importance of the aerosol height in interpreting the UVAI. Sun et al. (2018) explicitly

mention in their study the requirement of accurate aerosol layer height estimates in order to derive aerosol absorption from the UVAI.

The global monitoring of aerosol properties can only be done using remote sensing techniques from space. The space-based techniques currently utilised by the scientific community to retrieve aerosol vertical information are divided into two categories — active and passive techniques; active remote sensing techniques monitor aerosol properties by measuring the interaction of

energy generated by the instrument with aerosols in the atmosphere, whereas passive techniques do the same by measuring the interaction of natural light with aerosol particles. There are several differences in the sensing principles between active and passive remote sensing of aerosols, specifically in terms of vertical resolution. Active sensors such as the CALIOP instrument provide attenuated backscatter profiles resolved vertically at a vertical resolution as fine as 30 m for different channels over a spatial resolution as fine as 0.33 km (see Table 2 of Winker et al. (2009)). While these measured backscatter profiles provide

detailed quantitative information on the scattering species present in the atmosphere, spaceborne atmospheric profiling lidars have limited spatial coverage due to their limited beam width. Owing to this particular feature of active remote sensing, spaceborne lidars currently do not revisit a specific point on Earth several times a day, or even on a daily basis. On the other hand, passive spaceborne remote sensing has the ability to measure a specific point on Earth once a day for polar orbiting satellite missions and several times in the day for geostationary missions. Currently however, the retrieved information on

aerosol vertical distribution from passive remote sensing techniques is much more limited when compared to active techniques such as orbiting lidars.

Several passive retrieval strategies that are either currently in their operational phase or are upcoming remote sensing missions utilise the interaction of incoming solar radiation with the aerosol species to retrieve height information. Some notable mentions of missions that retrieve aerosol layer height are Multiangle Imaging Spectroradiometer (MISR) on board the NASA

Terra satellite (Nelson et al., 2013) which measures aerosol height using geometric optics, the Deep Space Climate Observatory (DSCOVR) mission with its Earth Polychromatic Imaging Camera (EPIC) (Xu et al., 2017, 2019), the Ozone Monitoring Instrument (OMI) on board the NASA Aura mission (Chimot et al., 2017, 2018; Choi et al., 2019), and finally the TROPOMI instrument on board the Sentinel-5 Precursor mission (Veefkind et al., 2012). Xu et al. (2017) and Xu et al. (2019) are the first study to demonstrate that the diurnal cycle of aerosol height is retrievable. In the near future, missions like the upcoming

Multi-Angle Imager for Aerosols (MAIA) mission (Davis et al., 2017), the Geostationary Environment Monitoring Spectrometer (GEMS) and the Tropospheric Emissions: Monitoring Pollution mission (TEMPO) are expected to provide aerosol height retrievals as well (Kim et al., 2018; Park et al., 2016; Zoogman et al., 2017). These instruments are examples of missions demonstrably (some theoretically, others practically) capable of retrieving aerosol layer height. Except for TROPOMI however, there ar currently no passive remote sensing mission that provides an operational stream of retrieved aerosol layer heights. In

the fourth quarter of 2019, an operational data stream of retrieved aerosol layer heights derived from measured oxygen A-band

spectra by TROPOMI has been made available to the general public; the TROPOMI operational UVAI product augmented by the TROPOMI ALH product has the potential to further the operational monitoring of aerosol properties globally. This paper discusses some key features of the product and its limitations by comparing it with co-located CALIOP profiles, and paints a future outlook of the evolution of the TROPOMI aerosol layer height algorithm. The paper looks into more than two million

colocations between TROPOMI ground pixels and CALIOP profiles over an extended period of time covering several months from May 2018 till March 2019, in order to draw conclusions on the accuracy of the TROPOMI aerosol layer height retrievals. Further on, the paper also discusses four selected cases in and around West Africa for a deeper analysis of the comparison with CALIOP data; the choice of using the Africa as a study area arises from the fact that a significant majority of colocations between TROPOMI and CALIOP are concentrated around the West African region.

In Section 2) of this paper, we discuss the data and methods used in this paper; section 2.1 describes the retrieval algorithm and highlights different diagnostic parameters available for assessing the product's quality. Following this, the comparison between CALIOP and TROPOMI estimates of aerosol heights are presented in 3 — Section 3.1 presents an overall analysis of a large number of TROPOMI-CALIOP colocations, followed by Section 3.2 which discusses selected cases for a deeper dive into the TROPOMI product. The paper concludes with section 4, highlighting important areas of potential improvement in the

current TROPOMI aerosol layer height product.

## 2   Data and Methods

### 2.1   TROPOMI aerosol layer height

The TROPOMI aerosol layer height product is derived from measurements of the oxygen A-band in the near infrared region between 758 nm and 770 nm. Within this spectral range, TROPOMI measures top of atmosphere radiances and solar irradiances

with a spectral resolution between 0.34 nm and 0.35 nm and a spectral sampling of 0.126 nm. The retrieval algorithm exploits the absorption characteristics of molecular oxygen, which varies with the photon path length — the photon path length for an aerosol layer closer to the surface is longer, which appears as deeper oxygen absorption lines in the measured spectrum (see Figure 1 of Nanda et al. (2018a)).

     The reported aerosol layer height is the height of a single aerosol layer for the entire atmospheric column within the scene

measured by TROPOMI; in reality however, there can be several cases where distinctly separated elevated and boundary layer aerosols are present in the same scene. In such cases, the retrieval algorithm is expected to retrieve an optical centroid pressure or height of the two (or more) aerosol layers, depending on the atmospheric level of the aerosol layer from which most of the photons are scattered back. For instance, if the elevated aerosol layer contributes significantly more than the boundary layer aerosols to the top of atmosphere measured spectra, the aerosol layer height retrieval algorithm is expected to retrieve values

closer to the elevated layer.

The technique for retrieving aerosol layer height is based on optimal estimation (Rodgers, 2000), where an RTM that calculates the top of atmosphere oxygen A-band spectra is fitted to TROPOMI measured oxygen A-band spectra. The cost function that is minimised in this estimation step, $\chi^2$, is defined as

$$\chi^2 = [\boldsymbol{y} - \boldsymbol{F}(\boldsymbol{x},\boldsymbol{b})]^T \mathbf{S}_\epsilon^{-1} [\boldsymbol{y} - \boldsymbol{F}(\boldsymbol{x},\boldsymbol{b})] + (\boldsymbol{x} - \boldsymbol{x_a})^T \mathbf{S_a}^{-1} (\boldsymbol{x} - \boldsymbol{x_a}), \tag{1}$$

where, $\boldsymbol{y}$ is the reflectance spectra calculated from measured radiances and irradiances for the oxygen A-band, $\boldsymbol{F}(\boldsymbol{x},\boldsymbol{b})$ is the modeled reflectance for input parameters $\boldsymbol{b}$, of which the state vector $\boldsymbol{x}$ containing aerosol layer height $z_{\mathrm{aer}}$ and aerosol optical thickness $\tau$ is a part, $\boldsymbol{x_a}$ is the a priori state vector and $\mathbf{S}_\epsilon^{-1}$ and $\mathbf{S_a}^{-1}$ are the measurement error covariance and the a priori error covariance matrices. Optimal estimation is an iterative process, requiring several iterations to minimise the cost function described in Equation 1. The approach is Gauss-Newton, with a maximum number of iterations set at 10. If the
optimal estimation does not converge within these iterations, the aerosol layer height field in the final level-2 product is filled with a fill value. For a given measurement, optimal estimation is said to have converged to a final solution if the update to the state vector for the next iteration is less than the expected precision.

The $\chi^2$ is a measure of how close the modelled sun-normalised radiances are to the observations, with smaller values representing a better fit. The consequence of the many assumptions in the model (described in Section 2.2 of Nanda et al.
(2019)) result in a large $\chi^2$ (to the order of 1E4-1E7), with larger $\chi^2$ representing a larger departure between the model and the observation. There are several reasons for these departures, the more important ones being the presence of undetected clouds in the scene, incorrect surface reflectance information, and multiple aerosol layers. These attributes are not parameterised into the RTM, and can be source of discrepancies between the measured and the modeled reflectances. The RTM in this case is a neural network model that has learned parts of a full physics RTM derived from de Haan et al. (1987), described in Nanda et al. (2019)
(Section 3), which is three orders of magnitude faster than DISAMAR. In short, the atmosphere is simplified by DISAMAR in order to reduce computational burden, and the neural network forward model is implemented for a further performance boost in an operational environment; for instance, DISAMAR ignores rotational raman scattering even though literature has shown that the oxygen A-band ring effects are sensitive to aerosol layer height (Vasilkov et al., 2013; Wagner et al., 2010). These decisions have been made in order to speed up line-by-line calculations of DISAMAR, which are the basis of the training data
for its neural network counterpart. This decision is motivated by preliminary sensitivity analyses conducted by Sanders and de Haan (2016) which conclude that the effect of ignoring RRS is not significant enough to venture in its implementation into the forward model.

The surface reflectance model used in the algorithm is derived from Tilstra et al. (2017), which is a Lambertian equivalent reflectance (LER) database with a spatial resolution of $0.25° \times 0.25$ °. In contrast to TROPOMI's ALH product which is
reported at 7.2 km $\times$ 3.6 km till August 6, 2019, and 5.6 km $\times$ 3.6 km thereafter, the LER database is much coarser spatially. This can lead to several artefacts in the final product, discussed further on in this paper in Section 3.2. Another issue to note is in the influence of bright surfaces on the retrieval. The oxygen A-band lies beyond the red edge, a wavelength region in which vegetation has high reflectance values. This poses several challenges; a significant portion of the measured signal over land

might be contributions from the surface reflectance (see Figure 3 from Nanda et al. (2018a)). If the aerosol optical thickness of the measured scene is low, the contribution of the surface to the top of atmosphere radiance dominates over the contribution from scattering by aerosols — there are more photons that get scattered back from the surface than the aerosol layer. In such cases, the retrieval algorithm will tend to retrieve an aerosol layer closer to the surface. Generally we find that, if the contribution to the top of atmosphere reflectance from aerosols is significantly larger than the same from the surface (i.e. the aerosol layer appears brighter than the surface), the retrieval algorithm will tend to retrieve a height closer to the aerosol layer (Section 5.2 and Figure 10 from Nanda et al. (2018b) discuss this observation explicitly).

The forward model parameterises aerosols with a Henyey-Greenstein scattering phase function (Henyey and Greenstein, 1941) with an asymmetry factor of 0.7, a single scattering albedo of 0.95, and a fixed aerosol optical thickness for an aerosol layer parameterised by a single atmospheric layer with a 50 hPa thickness. These assumptions have to be made since very little a priori information about aerosols in a scene is known. While more complex scattering models exist, the Henyey-Greenstein model has been used for retrieving ALH when the forward model was of line-by-line nature as the number of calculations it requires is far less than a scattering model such as the Mie model. Sensitivity analyses have shown that this assumption has few ramifications (Sanders et al., 2015). Fixing the single scattering albedo is a much bigger concern; while retrievals over the ocean do not suffer for a priori errors in the single scattering albedo, retrievals over land do have large errors and non-convergences which reduce as the the viewing zenith angle increases (Nanda et al., 2018a). The choice of using 0.95 as a fixed value arises from average values derived by Dubovik et al. (2002) from long-term observations using the aerosol robotic network (AERONET). The algorithm assumes a single aerosol layer for the entire atmosphere, within which aerosols are uniformly distributed and the aerosol volume extinction coefficient is constant. This is an important simplification to note when comparing with CALIOP profiles, since these lidar profiles have the capability to detect multiple aerosol layers. The simplicity in the aerosol profile parameterisation arises from the fact that it is impossible to know, without prior information, whether the scene consists of a single or multiple aerosol layers. While fitting of the aerosol layer pressure thickness along with the aerosol layer mid pressure does not result in large errors in the retrieved aerosol layer height, the precision of the retrieved aerosol layer mid pressure significantly deteriorates with increasing errors in the surface albedo (Sanders et al., 2015). More research has to be done before more information on the aerosol profile is retrieved from the oxygen A-band alone.

Finally, the ALH retrieval algorithm implements a pixel selection scheme before committing to retrieving ALH estimates. This pixel selection scheme involves auxiliary data products from TROPOMI such as the UVAI (www.tropomi.eu/document/atbd-uv-aerosol-index) and cloud fraction estimates from the TROPOMI Fast Retrieval Scheme for Clouds from Oxygen absorption bands (FRESCO) algorithm (Wang et al., 2008), and the cirrus reflectances derived from the Visible Infrared Imaging Radiometer Suite (VIIRS) on the Suomi National Polar-Orbiting Partnership (Suomi NPP) satellite.

1. The maximum solar zenith angle allowed is $75°$. If the pixel does not meet this criterion, it is removed from the processing and a flag is raised.

2. If the pixel over water lies in the sun-glint region (a maximum sun-glint angle of $18°$), it is processed but a sun glint warning flag is recorded in the level-2 product.

3. If the standard deviation of the surface elevation within the pixel is beyond 1000 m, the pixel is not processed and a flag is raised. If it is beyond 300 m, a warning flag is raised and the pixel is processed.

4. If the surface covered by the pixel comprises of both land and water, a warning indicating mixed surface type is raised and the pixel is processed regardless.

5. If the pixel contains snow or ice, the pixel is not processed and a flag is raised.

6. If the TROPOMI level-2 UV Absorbing Index product reports a value below 0.0, the pixel is not processed and a flag is raised. If the value is less than 1.0, a low UVAI flag is raised.

7. If the reported cloud fraction values from the TROPOMI FRESCO product for the pixel is beyond 0.6, the pixel is not processed and a flag is raised.

8. If the VIIRS average cirrus reflectance for the pixel is beyond 0.4, the pixel is not processed and a flag is raised. If it is beyond 0.01, a warning for possible cirrus clouds is indicated.

9. If the difference between the scene albedo (calculated using a look up table) from the Level-2 UVAI product and the surface albedo from the Tilstra et al. (2017) database at 380 nm is beyond 0.4, the pixel is removed from the processing pool and a flag is raised for possible cloud contamination. If this is value is beyond 0.2, a warning flag is raised.

10. The nominal TROPOMI pixels also contain radiances at a sub-pixel level, which are called small pixel radiances. If the standard deviation of the small pixel radiances is larger than 1E-7, the scene is deemed to be non-homogeneous (possibly containing clouds) and it is removed from the processing pool.

These relevant flags are reported in Table 1 and are available in the level-2 data products; the values for each of these flags can be accessed with bitwise-and operations for each pixel with the value of each processing quality flag. For cloud filtering, the cloud_warning flag is the preferred flag for removing possibly cloudy pixels. This flag is a combination of FRESCO cloud fraction retrievals, VIIRS cirrus reflectance retrievals and the difference between the surface albedo and the scene albedo at 380 nm. An example of applying the cloud_warning flags to filter out possibly cloudy pixels is provided in Figure 1.

## 2.2 CALIOP weighted extinction height

The Cloud-Aerosol Lidar with Orthogonal Polarisation (CALIOP) instrument is a part of the payload for the Cloud-Aerosol Lidar and Infrared Pathfinder Satellite Observation (CALIPSO) mission (Winker et al., 2009), which orbits the Earth in a sun-synchronous orbit. The CALIOP instrument has three backscatter receiver channel, two channels for the orthogonal measurement of received backscatter signal at 532 nm and one channel for backscatter at 1064 nm. Lidar profiles from the CALIPSO mission are a good source of data for validating retrieved aerosol layer heights from TROPOMI, because of their ability to map the vertical structure of the atmosphere. The data from the CALIOP instrument relevant for validating TROPOMI ALH are the level-1 backscatter profiles and the level-2 aerosol extinction profiles, which are used at the same time.

**Table 1.** Processing Quality Flags relevant for diagnosing S5P ALH product quality. The descriptions are derived from the S5P IODD.

| name | value | description |
|---|---|---|
| **CONVERGED PIXELS** | | |
| success | 0 | successful retrieval; warnings still possible. |
| sun_glint_warning | 2048 | pixel is in sun-glint region |
| cloud_warning | 32768 | combination of different cloud detection methods |
| UVAI_warning | 65536 | UVAI is lower than 1.0 |
| snow_ice_warning | 16384 | scene contains snow/ice |
| **NON-CONVERGED or MISSING PIXELS** | | |
| convergence_error | 19 | optimal estimation did not converge |
| sza_range_error | 7 | Solar zenith $\geq 75°$ |
| max_iteration_convergence_error | 21 | no convergence; retrieval exceeds maximum iterations |
| aot_lower_boundary_error | 22 | no convergence; AOT $\leq 0.0$ twice in succession |
| other_boundary_convergence_error | 23 | no convergence; state vector element crosses boundary conditions twice |
| solar_eclipse_filter | 64 | pixel not processed because of solar eclipse |
| cloud_filter | 65 | pixel skipped; FRESCO cloud fraction greater than 0.6 |
| altitude_roughness_filter | 67 | pixel skipped; STD of DEM in pixel $> 1000.0$ m |
| snow_ice_filter | 70 | pixel skipped; pixel contains snow/ice |
| UVAI_filter | 71 | pixel skipped; UVAI $< 0.0$ |
| cloud_fraction_fresco_filter | 72 | pixel skipped; cloud fraction $> 0.6$ |
| cirrus_reflectance_viirs_filter | 76 | pixel skipped; VIIRS cirrus reflectance $> 0.4$ |

In this paper, the level-1 total backscatter profiles from the 532 nm channel are used as curtain plots to visualise the vertical structure of the atmosphere. Level-2 aerosol extinction profiles from the 532 nm channel are then used to compute an aerosol weighted extinction height $\mathrm{ALH_{ext}}$, following the definition provided by Equation 1 in Koffi et al. (2012),

$$\mathrm{ALH_{ext}} = \frac{\sum_{i=1}^{n} \beta_{ext,i} Z_i}{\sum_{i=1}^{n} \beta_{ext,i}}, \qquad (2)$$

where $Z_i$ is the height from sea level in the $i^{\mathrm{th}}$ lidar vertical level $i$ (in km), and $\beta_{ext,i}$ is the aerosol extinction coefficient (in km$^{-1}$) at the same level. The Level-2 aerosol extinction profile product from CALIOP only includes atmospheric levels where aerosols are detected. In the case when aerosols are present over clouds, $\mathrm{ALH_{ext}}$ will be situated to the center of the aerosol layer, with any possibly undetected aerosol layers below the cloud layer not included in the calculations due to attenuation of the signal beyond the cloud layer. This is an important detail as the TROPOMI ALH algorithm cannot separate cloud

and aerosol signals from the measured radiances, and cloud contamination will affect the retrieved product. In this paper, the CALIOP 532 nm channel observations are chosen for analysis as the conclusions from the analysis of the results do not change when the 1064 nm channel observations are used instead. Appendix A explains the colocation technique used in this paper. The CALIOP aerosol product might be cloud contaminated as well, but this is difficult to ascertain. Plotting $\text{ALH}_{\text{ext}}$ over curtain

plots of level-1 total backscatter profiles can be used to visually discern possibly cloud-contaminated CALIOP level-2 aerosol product.

## 3  Results

### 3.1  Comparison of TROPOMI ALH and CALIOP $\text{ALH}_{\text{ext}}$

TROPOMI-CALIOP colocations between $1^{\text{st}}$ of May 2018 to the $28^{\text{th}}$ of February 2019 are selected. Two sets of overall

comparisons are done between CALIOP $\text{ALH}_{\text{ext}}$ and TROPOMI ALH, one with all colocations (Figure 2a) that aren't cloud filtered and the other with a smaller subset of the dataset constrained by the cloud_warning flag from Table 1 (Figure 2b). The contrast between retrievals over land and ocean is apparent in Figure 3 (cloudy scenes filtered out using the cloud_warning flag), with a majority of the negative differences with values lower than -2 km occurring over land.

From Figure 2a , what is immediately clear is that the CALIOP $\text{ALH}_{\text{ext}}$ are higher than the TROPOMI ALH. With an

average difference of -2.25 km, median difference of -1.62 km and a standard deviation of 3.83 km, the retrieved ALH from TROPOMI over land is reported to be systematically closer to the surface than CALIOP $\text{ALH}_{\text{ext}}$ than in comparison with retrievals over the ocean, which has a mean difference of -0.41 km, a median difference of -0.29 km and a very high standard deviation of 6.86 km. There are several cases over the ocean where TROPOMI ALH is significantly higher than the CALIOP $\text{ALH}_{\text{ext}}$, which could be due to cloud contamination. The comparison of the cloud-screened retrievals (Figure 2b) reveals that

the retrieved ALH from TROPOMI over the ocean differs from CALIOP $\text{ALH}_{\text{ext}}$ by -1.03 km on average, a median difference of -0.76 km and a standard deviation of 1.97 km. More than 50% of the TROPOMI ALH retrievals over the ocean have an absolute difference with $\text{ALH}_{\text{ext}}$ less than 1.0 km. Retrievals over land are have a larger difference, with -2.41 km on average and a median of -1.75 km. The results are very skewed over land, with very large negative values dictating the average — this is indicated by the very large standard deviation of 3.56 km. 50% of the selected colocations over land have an absolute

difference with $\text{ALH}_{\text{ext}}$ less than approximately 1.8 km.

The distribution of the differences between TROPOMI ALH and CALIOP $\text{ALH}_{\text{ext}}$ as a function of the retrieved UVAI (Figure 4a) shows that for most cases, the UVAI is below 2.0. The spread of the differences in this UVAI regime is large, which reduces as the UVAI increases. The differences seem to be less often positive as the UVAI increases; if compared with the behaviour observed between Figure 2a and Figure 2b where a majority of the positive differences vanish once the data is

cloud screened, such a behaviour could be related to clouds. The distribution of the differences as a function of retrieved AOT in Figure 4b show that the majority of the colocations have AOT values between 0 and 2. Finally, the distribution of these differences as a function of the GOME-2 LER values used for the retrievals for cases over land show that the retrievals tend to have a lower difference as the LER value increases — this could be a consequence of the fact that so few retrievals converge

in high LER regimes that, unless the aerosol layer has a significant contribution to the measured top of atmosphere radiance in comparison to the surface, the retrievals tend to fail.

Retrieved ALH over land (if successful) can be closer to the surface than where the aerosol layer actually is situated vertically. The TROPOMI ALH product, unlike the CALIOP $ALH_{ext}$ which only considers aerosol signatures in the recorded
backscatter profile, is also influenced by the presence of undetected clouds. These are some of the several possible sources of departures between the observations of CALIOP and TROPOMI over the same scene. In the next section, the comparison between TROPOMI ALH and CALIOP $ALH_{ext}$ is extended for a few selected scenes.

## 3.2   Analysis of selected cases

### 3.2.1   Selected cases

The analysis presented in the previous section alone is insufficient to fully quantify the quality of the retrieved TROPOMI aerosol layer heights, due to the manner in which clouds are handled by both aerosol heights; TROPOMI pixels are affected by the presence of undetected clouds whereas CALIOP aerosol extinction profiles do not consider clouds. Another significant source of departure between TROPOMI and CALIOP is their differing sensing principles. Making conclusions on the quality of the current TROPOMI ALH product requires case-by-case studies of selected scenes. In line with this, four cases are selected
to represent a very good mix of scenes containing elevated aerosol layers as well as aerosol layers close to the surface, high and low UV absorbing index, clear and cloudy scenes, clouds over and below aerosol layers, multiple aerosol layers, and retrievals over land and the ocean.

The cases selected are Saharan desert dust and biomass burning events, three off the west coast of Sahara (desert dust) in June 2018 and one off the South Saharan coast (biomass burning) in December 2018. All four cases have very good colocations
between TROPOMI and CALIOP, with the CALIOP ground track over the aerosol plumes (plotted with a yellow line over the VIIRS images in Figure 5 (1[st] column). The operational ALH level-2 algorithm operates on pixels falling within the sunglint regime, however they are excluded from the analyses presented in this paper. The retrieved UV absorbing index (UVAI) from the operational level-2 UVAI product gives an idea about the shape of the aerosol plumes in all these cases (Figure 5 (2[nd] column)). The UVAI is influenced by many factors including the height of the aerosol layer, with lower UVAI values
for aerosol layers closer to the surface (discussed further in Appendix B). Cases a and b contain several pixels with UVAI values greater than 3.0, whereas a majority of the TROPOMI pixels in cases c and d have TROPOMI UVAI values between 0.0 and 2.0. A significant majority of successful the retrievals in these selected scenes are over a dark surface, owing to the bright surface albedo of the Saharan desert. The reader is point to Griffin et al. (2019) for comparison of the TROPOMI ALH retrievals over land for biomass burning aerosol plumes with the same from several other instruments including CALIOP.

It is important to note that spaceborne lidars, while having the advantage of being able to map more than one vertical layer in the atmosphere, suffer from attenuation of the signal in the presence of strongly backscattering species such as clouds or aerosols with a large optical depth. In the presence of a primary strongly backscattering aerosol layer, the attenuation of the signal may lead to undetected secondary aerosol layers beneath the primary layer. These layers, not apparent in the CALIOP

curtain plots of the measured attenuated backscatter profiles, may be detected by the level 2 aerosol extinction profile product from the CALIOP mission, using the formula described in Equation 2. Some of these discussed situations are observed in the CALIOP curtain plots of the selected cases in Figure 6, especially for cases a and b, where the attenuated signal does not detect possibly lower aerosol or cloud layers, and in case d where the attenuation of the signal due to a thick aerosol plume can hide the surface from the received backscatter signal. TROPOMI, on the other hand, will tend to report an aerosol layer height between these two layers as it will be influenced by photons scattered back from both layers.

### 3.2.2 Analysis

The retrieved TROPOMI ALH in Figure 5 (4$^{\text{th}}$ column) represent successful retrievals for each of the selected cases. Beyond the sun glint warning, the cloud_warning flag in Table 1 is applied to remove possibly cloud contaminated data. The retrieved aerosol optical thickness (AOT), which is a part of the state vector, for each of the scenes are plotted over the VIIRS image of the scene in Figure 5 (3$^{\text{rd}}$ column). The retrieved AOT ($\tau_{\text{aer}}$) can act as a diagnostic tool to indicate the influence of the surface (over bright surfaces) or the presence of undetected clouds (both over bright and dark surfaces) — in these cases, the retrieved AOT of the scene can be uncharacteristically high with values much greater than 3.0. All retrieved TROPOMI AOT values beyond 5.0 are discarded as the neural network forward models are trained with AOT values less than or equal to 5.0.

A visual inspection of the figures in Figure 5 shows that the retrieved UVAI, AOT and ALH need not be spatially correlated, as they are separate properties of the observed aerosol plumes — for instance, if the retrieved UVAI and AOT are low (case c), the retrieved ALH need not necessarily be low. An inspection of the plots of the retrieved AOT for cases c (between latitudes 10° and 15° and longitude -20°) and d reveal square structures, both over the ocean and land. These square shaped spatial artefacts are the surface albedo grids derived from the database provided by Tilstra et al. (2017), which is the current source for surface reflectance in the ALH retrieval algorithm. In cases such as case c, the retrieved AOT contains surface information influenced by the assumed albedo in the database. These spatial features are not as apparent in cases a and b (Figure 5, 1$^{\text{st}}$ and 2$^{\text{nd}}$ rows) as a majority of the signal in the measured top of atmosphere radiances come from aerosols and the minority from the surface. Another major observation is the lack of retrievals over the desert. This is within expectation, as measurements of the top of atmosphere radiances over a cloud-free desert scene tend to contain more photons scattered back from the surface than the aerosol layer. As a result, retrievals over bright scenes are sensitive to the assumed errors in the surface albedo, thereby reducing sensitivity to the assumed aerosol layer height (Sanders et al. (2015), Section 2, Figure 2).

While scenes not contaminated with clouds show a smooth spatial distribution of the retrieved ALH, the presence of clouds may or may not add spatial variability in the ALH product. For instance, the the presence of low clouds are clear in case b (Figure 5b) beyond latitude 21.0°, but the retrieved ALH is spatially homogeneous with values less than 1.0 km. For each of the selected cases, colocated CALIOP profiles in Figure 6 give additional information about the scene. These TROPOMI-CALIOP colocations are done via the method discussed in Appendix A. The CALIOP curtain plot for case b reveals the influence of low clouds as well as high clouds on the cloud-screened ALH. An example of cloud-contaminated heterogeneous vertical distribution of TROPOMI ALH in Figure 6a can be observed between latitudes 9.5° and 11.0°. The cloud filtering following

the cloud_warning flag in Table 1 does not detect these low clouds (for instance above latitude 21.50°, see Figure 6 a, b). These are manually for comparison further on.

From Figure 2b, TROPOMI retrievals of ALH over bright surfaces are expected to differ from CALIOP $ALH_{ext}$, wherein the TROPOMI ALH product may report ALH estimates closer to the surface than CALIOP will. This is observed in case d (Figure 5, bottom row), wherein the CALIOP curtain plot for (Figure 6d) indicates that the plume is close to the surface, with a maximum height less than 3 km; TROPOMI ALH for biomass burning aerosol plume that extends from land to the ocean is slightly closer to the surface in the case of land when compared to CALIOP $ALH_{ext}$, whereas over the ocean both height estimates more or less are in agreement.

For cases a and b, retrieved TROPOMI ALH does not seem to coincide with large values of the received backscatter signal in the level-1 data, whereas it does for case c, and to a certain extent for case d (over land it tends to be closer to the surface). Parts of the CALIOP curtain plots for cases a, b and c suggest that a possible second layer beneath the layer that is visually obvious, or that the desert dust layer extends deeper to the surface and the CALIOP signal is simply too attenuated to detect it.

A direct comparison of the CALIOP $ALH_{ext}$ and TROPOMI ALH for these four selected cases are presented in Figure 7. For this comparison, every cloud-filtered and sun-glint-filtered TROPOMI pixel with ALH information colocated to a specific CALIOP level-2 aerosol extinction profile in Figure 6 is averaged and a standard deviation is also computed. These averaged TROPOMI ALH are then compared to the CALIOP $ALH_{ext}$, and show that TROPOMI ALH differ from CALIOP $ALH_{ext}$ by 0.53 km, with a pearson correlation coefficient of 0.64 and a slope of 1.0; CALIOP $ALH_{ext}$ are systematically higher than TROPOMI ALH (indicated by a y-intercept of the fit at 0.53 km). The CALIOP $ALH_{ext}$ is also higher than TROPOMI ALH almost consistently in most cases. This could possibly be due to CALIOP possibly underestimating the aerosol layer thickness due to strong attenuation of the lidar signal at the top of the aerosol layer (Rajapakshe et al., 2017), whereas TROPOMI ALH product does not suffer from such attenuation.

## 4  Discussion and conclusion

This paper discusses the quality of the soon to be released TROPOMI ALH product by comparing it with CALIOP data of colocated measurements of scenes containing aerosols between the two instruments. In order to do so, CALIOP weighted extinction heights from the 532 nm channel were calculated following Equation 2, and then directly compared to TROPOMI ALH. Further on, four individual cases of Saharan desert dust and biomass burning aerosol events in 2018 were selected for a deeper analysis of the product's quality.

From the analysis presented in this paper, TROPOMI's neural network ALH retrieval algorithm retrieves ALH values that compare well with CALIOP weighted extinction heights in cloud-screened cases following the cloud screening strategy using the TROPOMI ALH level-2 processing quality flags discussed in Table 1. For more than 1 million colocations between CALIOP and TROPOMI over the ocean, the TROPOMI ALH differs from CALIOP $ALH_{ext}$ on average by approximately -1 km and -0.76 km median, with the TROPOMI ALH values being lower than the CALIOP $ALH_{ext}$. Over land, the same values are -2.41 km on average and -1.75 km as the median. To get a better understanding of the differences between TROPOMI and

CALIOP retrieved aerosol heights, this paper compared the aerosol layer height for selected cases among the more 1 million colocations that were better understood. The four selected scenes were chosen around the West Saharan region, where a majority of the TROPOMI-CALIOP colocations were found. For the selected cases, largely over the ocean within a portion of the data over land, the averaged retrieved ALH from TROPOMI differed from CALIOP $ALH_{ext}$ by 0.53 km, with CALIOP $ALH_{ext}$ being higher than TROPOMI ALH. These numbers are indicative that TROPOMI ALH performs well, especially considering the many simplifications made by the retrieval algorithm in order to optimise on the computational speed; future improvements to the forward model may only improve the product further on.

There is a clear distinction between TROPOMI ALH retrievals over land and the ocean as photons scattering back from bright surfaces tend to influence ALH estimates closer towards the surface than an elevated aerosol layer. Retrieved ALH over land, if successful, can to be closer to the surface if measured signal in the top of atmosphere contains more photons scattered back from the deepest atmospheric layer which is the surface, in comparison to elevated aerosol layers which are higher up in the atmosphere. This, however can change depending on the amount of aerosol information available in the spectrum compared to same from the surface. Any attempt in retrieving ALH over the desert generally fail, with very few exceptions. There are several challenges, that will need further development.

The TROPOMI level-2 UVAI product is currently an ingredient in selecting pixels containing aerosols for retrieving ALH. While this choice works quite well for cloud free scenarios, it does not do a great job when a scene that contains both aerosols and clouds. These cloudy scenes seem to not be detected by the current cloud filtering schematic in the level-2 algorithm, and will require a significant update in deciding whether a pixel is cloudy or not. For cases scenes with a low aerosol load, square shaped artefacts resulting from a surface albedo database with a resolution significantly lower than TROPOMI exist. Currently, the GOME-2 surface LER product derived from Tilstra et al. (2017) is used operationally, and will eventually need to be updated with a higher resolution version possibly derived from TROPOMI itself. To that extent, owing to the boost in the computational speed of the radiative transfer calculations, the algorithm can now incorporate more complex aerosol property and profile parameterizations. Such a step will benefit the TROPOMI aerosol layer height retrieval accuracy significantly.

Finally, space based lidars such as the CALIOP instrument on board the CALIPSO mission are a very good source of aerosol vertical information to validate the TROPOMI ALH product. While the CALIOP level-1 backscatter profiles may be attenuated in cases of very strong signals from the top of the aerosol layer, the weighted extinction heights in conjunction with the backscatter profiles are sufficient for validation activities. These CALIOP profiles will be very important in assessing the impact of future development activities of the TROPOMI ALH product.

**Appendix A: Colocation**

The colocation between TROPOMI and CALIOP ground pixels is done in the following manner. First, the geographic coordinates of CALIOP level 1 backscatter profiles and level 2 aerosol extinction profiles are converted into the Cartesian coordinate system. These CALIOP coordinates are fed into a k-dimensional tree, which is a fast algorithm developed by Maneewongvatana and Mount (1999) to quickly locate the nearest neighbour of a point (a TROPOMI ground pixel) to a k-dimensional tree

of points (CALIOP ground pixels). The scipy.spatial.KDtree module in python3 is used to create the k-dimensional tree of the ground coordinates of CALIOP profiles (separate for level 1 and level 2 data). Second, all TROPOMI ground pixel coordinates are converted to Cartesian coordinates. For each of these TROPOMI pixels, the distance to the nearest CALIOP profile is queried using the scipy.spatial.KDtree.query function. This creates a list of TROPOMI pixels and their nearest CALIOP profile and a distance in meters. Finally, only co-locations with a maximum distance of 100 km and a maximum time difference of 5 hours are selected. A map of all 2,474,042 colocations (in Figure 8) shows that most of the colocations are close to the continent of Africa. After filtering out all colocations in the TROPOMI sun-glint region, all retrieved aerosol optical thicknesses greater than 5.0 (as the neural network is trained for all AOT less than 5.0), and filtering out ocean pixels with a surface albedo greater than 0.05 and land pixels with a surface albedo less than 0.1 and greater than 0.4, there are in total 731,347 TROPOMI pixels entirely over land and 1,742,695 pixels entirely over water (see Figure 2a). After cloud screening using the cloud_warning flag in Table 1, a total of 546,445 pixels over land and 1,036,550 pixels over the ocean remain (see Figure 2b).

## Appendix B: UVAI Sensitivity to aerosol layer height

It is well-documented that the UVAI depends on aerosol layer height (Herman et al., 1997; Torres et al., 1998; de Graaf, 2005; Sun et al., 2018). Absorbing aerosols mainly interact with molecular scattered radiation beneath the aerosol layer. The higher the layer, the more Rayleigh scattering underneath is shielded, leading to a high UVAI value (Figure 9a). This altitude dependence increases with aerosol absorptions (i.e. SSA) and aerosol loading (i.e. AOD), whereas it becomes weaker over brighter surfaces where the importance of molecular scattering reduces significantly (Figure 9b). On the other hand, little altitude dependence is found for non-absorbing aerosols (i.e. SSA = 0.99). The conclusions from this synthetic experiment are replicate with real TROPOMI data in a separate manuscript, where for retrieved ALH for pixels with a UVAI greater than 1 for measurements from TROPOMI showed an increase in the correlation as well as the slope between ALH and UVAI for an increase in MODIS aerosol optical depth values for the same scenes. This manuscript is currently submitted to Atmospheric Chemistry and Physics and awaits review.

*Competing interests.* The author declares no conflict of interests in the work expressed in this publication.

*Acknowledgements.* This publication contains modified Copernicus Sentinel data. This research is partly funded by the European Space Agency (ESA) within the EU Copernicus programme. We acknowledge the use of VIIRS imagery from the NASA Worldview application (https://worldview.earthdata.nasa.gov/), part of the NASA Earth Observing System Data and Information System (EOSDIS).

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

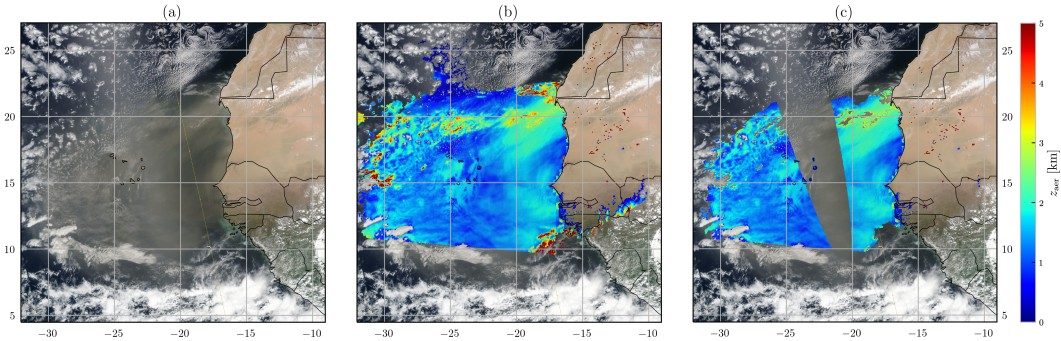

**Figure 1.** **(a)** A VIIRS corrected reflectance image over the West African coast on the 8<sup>th</sup> of June, 2018. **(b)** All successful TROPOMI retrievals within a certain bounding box. **(c)** Same as **(b)** but with all pixels that possibly fall within the sun glint region or are cloud contaminated are removed (using cloud_warning flag and sun_glint_warning from Table 1).

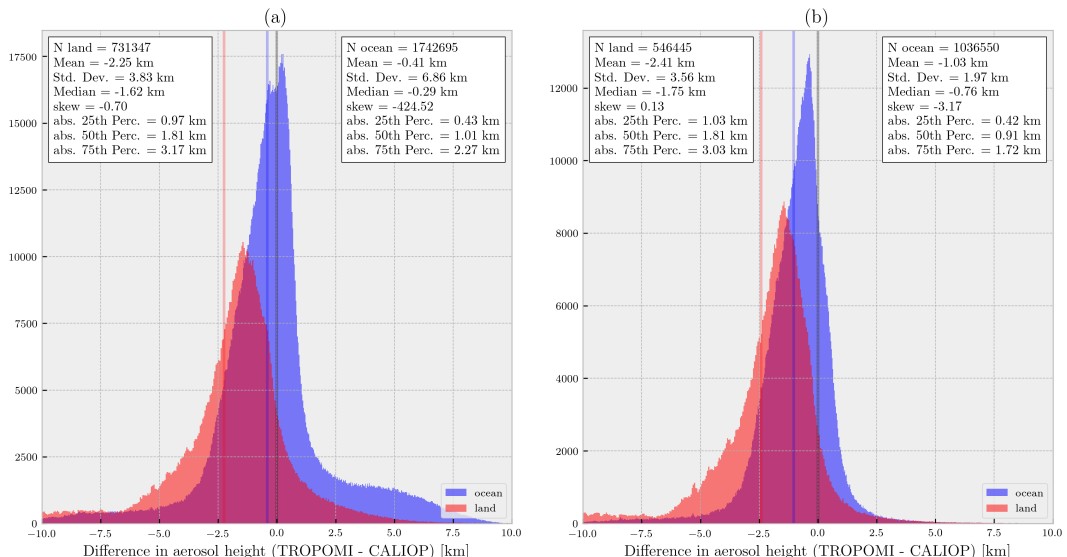

**Figure 2.** Histogram of differences between CALIOP $ALH_{ext}$ (Equation 2) and TROPOMI ALH from colocated data between May 1, 2018 and February 28, 2019. Blue histogram represents TROPOMI pixels over the ocean whereas the red histogram is for TROPOMI pixels over land. The blue line represents the mean difference between TROPOMI ALH and CALIOP $ALH_{ext}$ for TROPOMI pixels over the ocean, whereas the red line represents the same for TROPOMI pixels over land. The black line at 0.0 km difference on the x-axis is plotted to aid the reader in their interpretation of this figure. **(a)** All colocations except TROPOMI pixels falling in the sun glint region. TROPOMI pixels with retrievd AOT greater than 5.0 are discarded. For pixels over land, if the GOME-2 surface albedo is less than 0.1 or greater than 0.4, they are discarded. Similarly, over the ocean all TROPOMI pixels that have a GOME-2 surface albedo greater than 0.05 are discarded. **(b)** Same, except only TROPOMI ALH retrievals that are cloud-screened using cloud_warning flag from Table 1 are included.

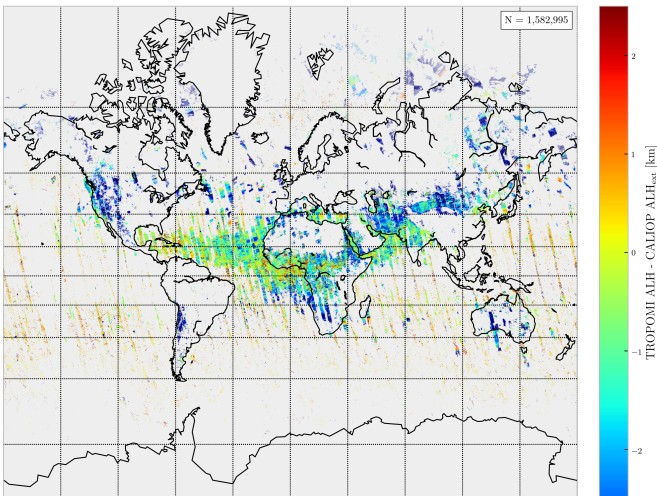

**Figure 3.** A map of cloud filtered and sun glint filtered differences between colocated TROPOMI ALH and CALIOP ALH$_{ext}$ considered for Figure 2b.

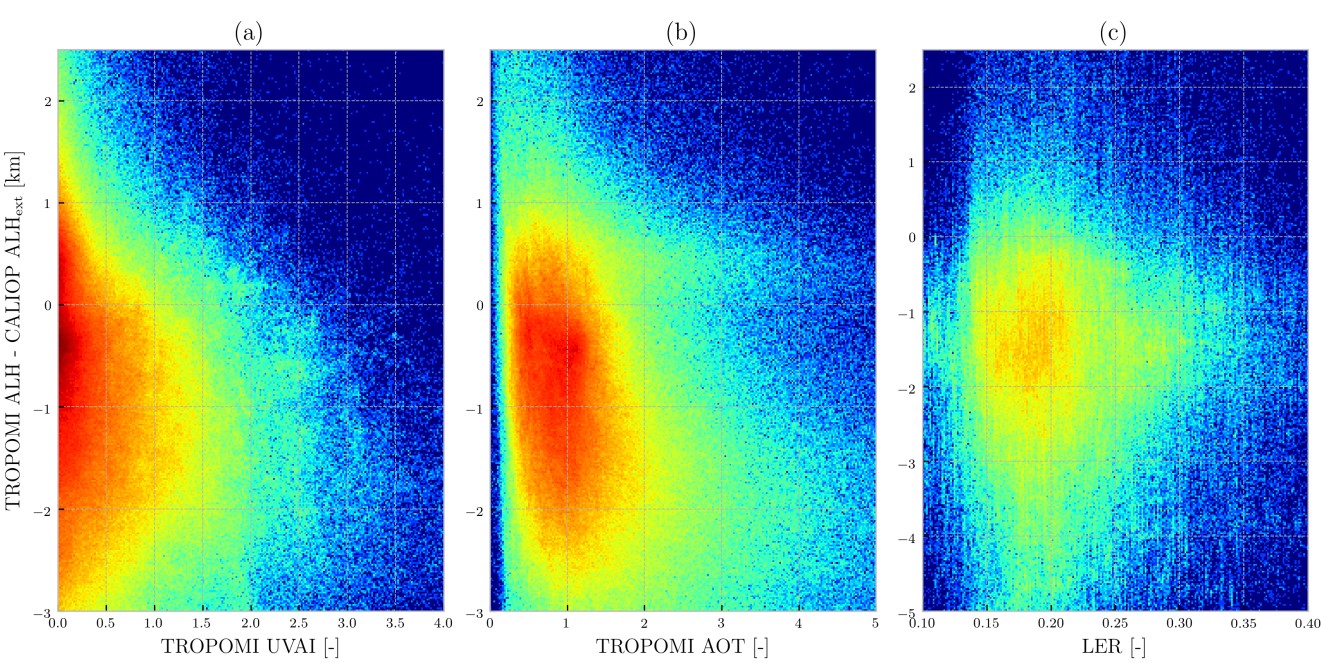

**Figure 4.** Scatter density plots of the difference between TROPOMI ALH and CALIOP ALH$_{ext}$ as a function of **(a)** TROPOMI UVAI, **(b)** TROPOMI AOT and **(c)** GOME-2 LER for the oxygen A-band used for the TROPOMI retrievals for cases over land (with a minimum surface albedo of 0.1). The colors represent density of plots. The y axis is optimised for each plot. The data is filtered in the same fashion as in Figure 2, with data over the ocean and land combined for plots **(a)** and **(b)**, and data only over land for plot **(c)**.

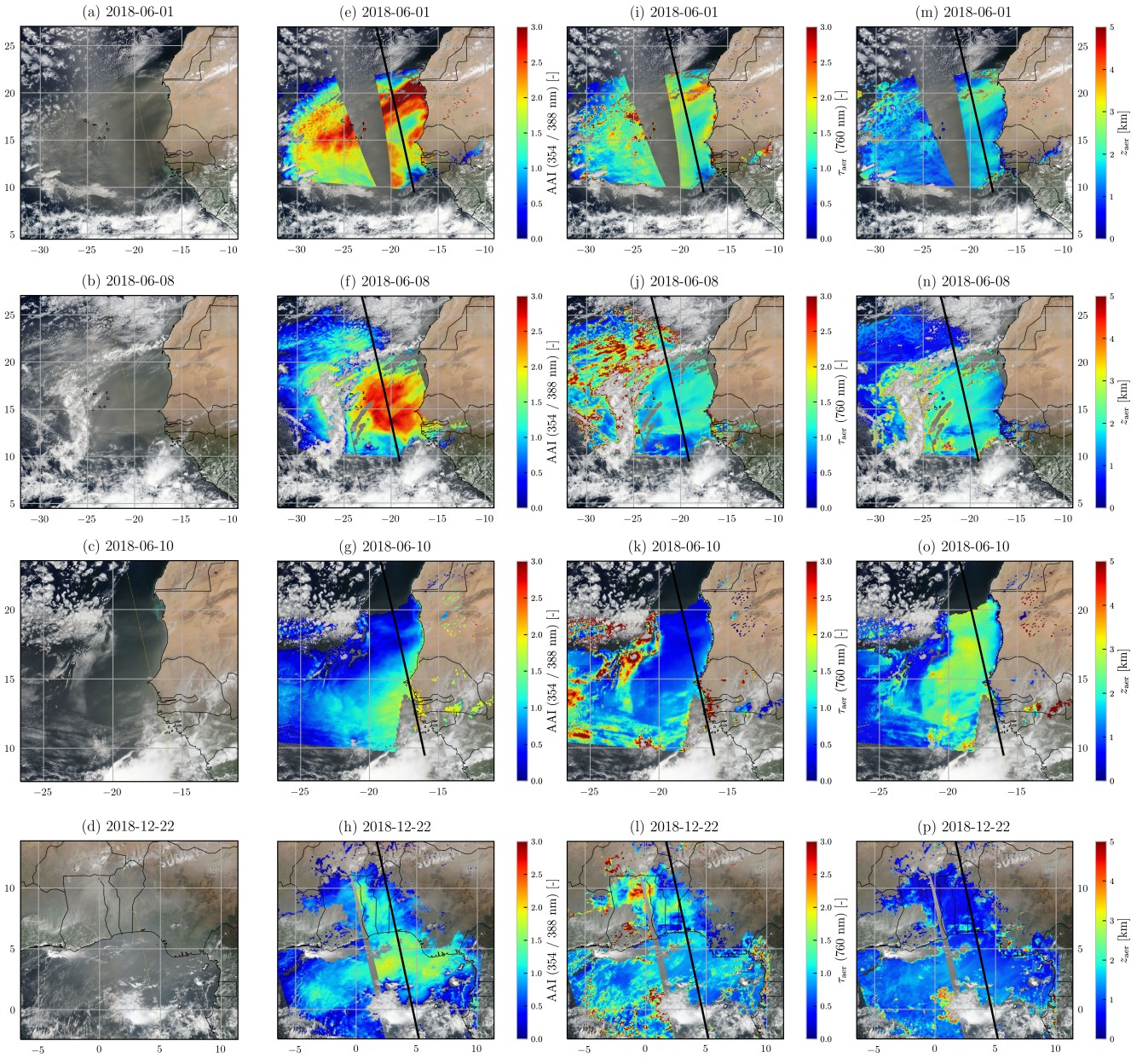

**Figure 5. 1st column:** Corrected reflectance for the four selected cases as measured by the Suomi NPP/VIIRS imager. The yellow line represents the CALIOP ground track. **2nd column:** The TROPOMI level-2 UV Absorbing Index product. The black line passing through the TROPOMI level-2 retrievals on this plot represents the ground track of the CALIPSO mission. **3rd column:** Retrieved aerosol optical thickness from TROPOMI. **4th column:** Operational TROPOMI aerosol layer height.

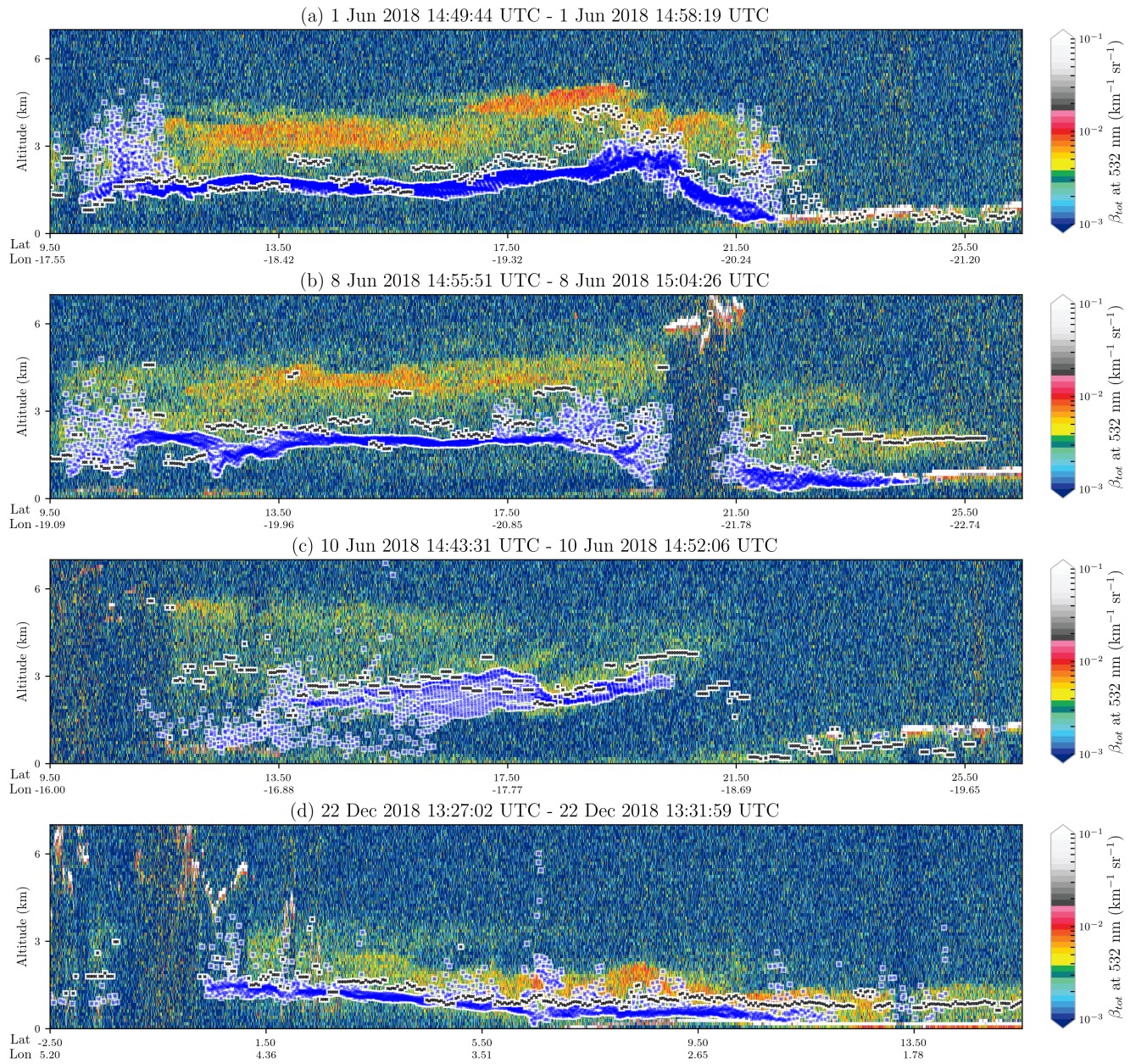

**Figure 6.** CALIOP level-1 backscatter curtain plots for measurements in the 532 nm channel for the four selected cases in Figure 5. The blue markers (crosses over a white box) represent co-located TROPOMI ALH retrievals within 100 km of each CALIOP profile present in this plot. The black markers (crosses over a white box) represent the CALIOP weight aerosol heights as computed using Equation 2. TROPOMI data that are either in sun-glint region or cloud contaminated are removed (cloud detection is done using the cloud_warning flag from Table 1).

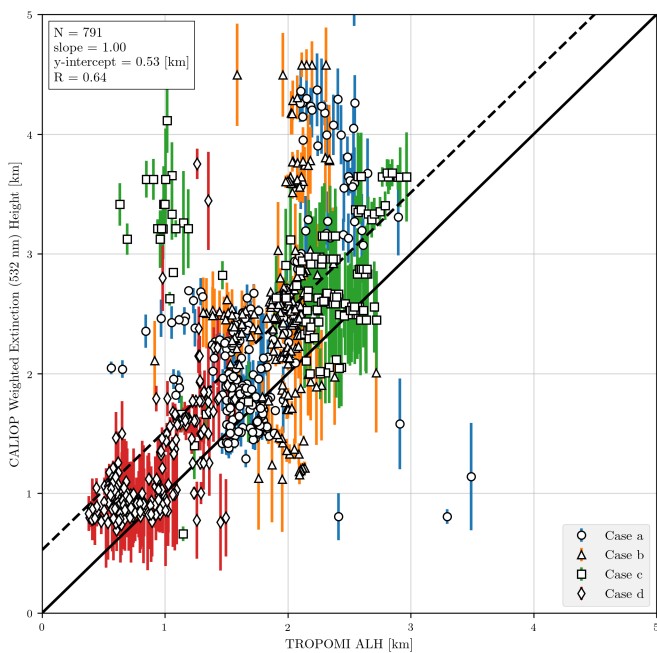

**Figure 7.** Comparison between the CALIOP weighted extinction heights (y-axis) calculated using Equation 2 and plotted in Figure 6, against averaged TROPOMI ALH (x-axis). The blue lines represent the standard deviation of the TROPOMI heights in the averaging pool, and the markers represent the mean TROPOMI ALH for each CALIOP $\text{ALH}_{\text{ext}}$. The dashed black line marks the fit between CALIOP $\text{ALH}_{\text{ext}}$ and TROPOMI ALH. The solid black line is a neutral line between the x and the y axes. The legend in the bottom right corner describes the different markers used for the different cases.

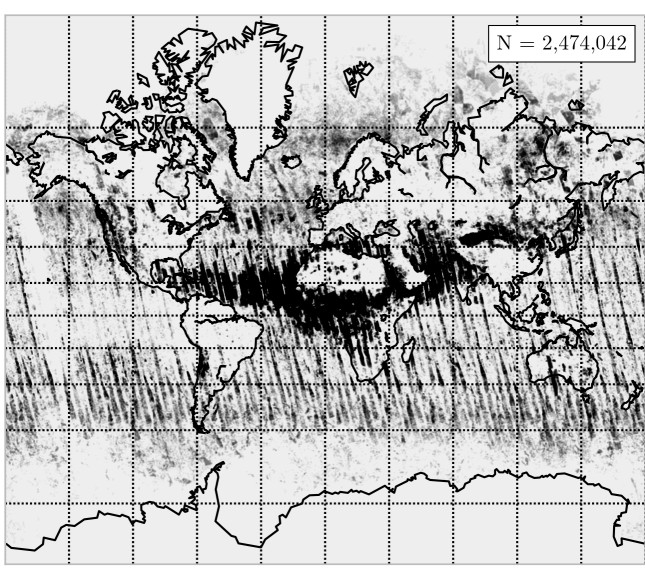

**Figure 8.** A map of all TROPOMI-CALIOP colocations considered for Figure 2 (data filtering discussed in Appendix A).

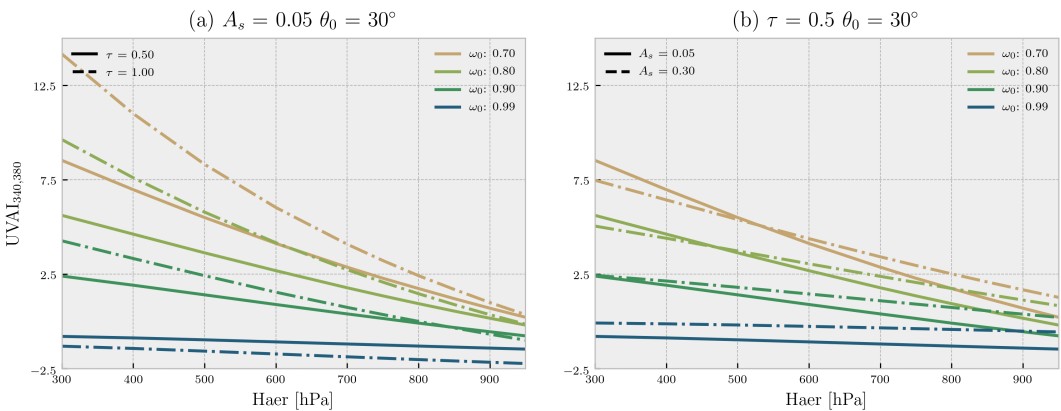

**Figure 9.** Sensitivity analysis of UV aerosol index to show the influence of different aerosol properties on the UVAI. The aerosols in these analyses have a Henyey-Greenstein scattering phase function with an asymmetry factor of 0.7, an angstrom exponent of 1.0, the viewing zenith angle is $0°$, the solar azimuth angle and the viewing azimuth angles are $0°$ and $60°$ respectively, the surface pressure is 1013 hPa, and for this specific case, the solar zenith angle $\theta_0$ is $30°$. The y-axis is the UVAI for 340 nm and 380 nm, whereas the x-axis is the height of the geometric centroid of the aerosol layer in hPa (Haer). The legend in each of the plots describe the different configurations chosen for these sensitivity analyses. **(a)** looks into the sensitivity of UVAI with a fixed surface albedo of 0.05, and **(b)** does the same for a fixed aerosol optical thickness of 0.5.