# Peer review of "A first comparison of TROPOMI aerosol layer height to CALIOP data"

_Atmospheric Measurement Techniques, 2019_

## Referee Comment (RC1) · Anonymous Referee #1 · 26 Dec 2019

The manuscript entitled by "Validating TROPOMI aerosol layer height retrievals with CALIOP data" shows the initial validation results of aerosol vertical structure information from TROPOMI sensor. The aerosol vertical information is important result for the trace gas retrieval and air quality information relating to the PM2.5 etc. For this reason, the validation of aerosol layer height retrieval result is essential to publish. However, several supplements are required before the publication.

Specific comments 1) In page 2, Lines 23: Although the aerosol layer information by the environment satellite mission is limited, several previous studies were investigated including sensitivity results and methodology. Therefore, please add the reference for the aerosol height retrieval algorithm relating to next generation of environmental satellites (such as GEMS, TEMPO etc.).

e.g.) Choi, Wonei, et al. "Effects of spatiotemporal O4 column densities and temperature-dependent O4 absorption cross-section on an aerosol effective height retrieval algorithm using the O4 air mass factor from the ozone monitoring instrument." Remote Sensing of Environment 229 (2019): 223-233. Kim, Mijin, et al. "Optimal Estimation-Based Algorithm to Retrieve Aerosol Optical Properties for GEMS Measurements over Asia." Remote Sensing 10.2 (2018): 162. Park, Sang Seo, et al. "Utilization of O 4 slant column density to derive aerosol layer height from a space-borne UV–visible hyperspectral sensor: sensitivity and case study." Atmospheric Chemistry and Physics 16.4 (2016): 1987-2006. Zoogman, P., et al. "Tropospheric emissions: Monitoring of pollution (TEMPO)." Journal of Quantitative Spectroscopy and Radiative Transfer 186 (2017): 17-39. Vasilkov, A., J. Joiner, and R. Spurr. "Note on rotational-Raman scattering in the O 2 A-and B-bands." Atmospheric Measurement Techniques 6.4 (2013): 981-990. Wagner, T., et al. "A sensitivity analysis of Ring effect to aerosol properties and comparison to satellite observations." Atmospheric Measurement Techniques 3.6 (2010): 1723-1751.

2) In page 4, Lines 34 : For the forward model simulation, the aerosol optical and physical properties based on the Henyey-Greenstein scattering phase function is insufficient. Also, the fixed single scattering albedo affects the estimation errors due to the variability of aerosol optical properties. The atmospheric layer is also too simple as we compared to the previous researches of aerosol height estimation studies. Author has to be explained the reason of simple assumption for aerosol optical and physical properties in the TROPOMI algorithm. Especially, retrieval error of aerosol height relating to the single scattering albedo and size information were reported in several previous studies.

3) In page 6, line 11: For the validation of ALH, author used both level 1 and 2 data of CALIPSO. If both data exists, which of the two data do you use first?

4) In page 7: The CALIOP data has potential error to classify cloud and aerosol. For the validation, additional consideration for cloud contamination in the aerosol products

of CALIOP is also important.

5) Figure 2: From the Fishman et al. (2012) in BAMS, the reference value of aerosol layer height error is 1 km. However, only 50% of the data satisfies the error within 1 km, and the standard deviation is always larger than 1 km as author wrote in the manuscript. Compared to the expected error (1 km), the error is relatively large. Given these results, do you think the accuracy of these results is sufficient?

6) Figure 7: Compared to the slope value, the Y-intercept is too large. Please discuss the reason of large positive bias of Y-intercept.

7) In Page 11: For the further study, the author discusses to update the LER product. However, updating aerosol properties are most important point in this study. Please add the author's opinion.

Simple Technical comment:

1) In page 9 (Line 17) : correct the typo-error (4thrd -> 4th)
* * *

---

## Referee Comment (RC2) · Anonymous Referee #2 · 11 Jan 2020

The paper presents the first attempt to validate the TROPOMI ALH data product and
therefore, bears lots of research interest from the community. The work overall is
sound. However, given the unprecedented observation from TROPOMI and its po-
tential for the wide use of ALH, the paper should be revised to provide an uncertainty
estimate to the community. In fact, from what is presented in the paper, the ALH at
the pixel level appears to have large uncertainties. Perhaps changing 'validating' to
'first evaluating" is more appropriate for this paper – this is a suggestion though. Major
recommendations are provided below.

1) While the introduction part mentioned several papers regarding ALH retrieval, it
didn't go in depth to the method themselves. A few notes are highlighted below; more
details can be found in Xu et al. (2017).

a. MSIR offers stereo height information; this is simply done by geometric optics, and providing top height of the aerosol layer

b. Xu et al., 2017: used primarily O2 A band (I think) for ocean, while Xu et al. 2019 used O2 B band primarily over land. In both cases, these two papers demonstrate for the first time in the literature that diurnal variation of ALH can be retrieved. They are also the first to define the method to evaluate ALH from such retrievals. Somewhere, it is worthy mentioning these, for example, in the method part for equation 2, and in the analysis and discussion part regarding surface reflectance.

c. MAIA by Davis et al. This is really a theoretical work as MAIA is not launched yet. This work should be separated from the work that uses the real data, and should be lumped with other theoretical work (such as Ding et al., 2017).

Reference: Xu et al., (2018), Passive remote sensing of aerosol height, in Remote Sensing of Aerosols, Clouds, and Precipitation, edited by T. Islam, Y. Hu, A. Kokhanovsky, and J. Wang, pp.1-22, Elsevier, Cambridge, MA.

Ding et al. (2016), Polarimetric remote sensing in O2 A and B bands: Sensitivity study and information content analysis for vertical profile of aerosols, AMT

2) The analysis part is really short in this paper. A few questions are suggested here with a hope to improve the analysis and add more 'meat' to the paper.

a. It is worth mentioning that aerosols, unlike cloud droplets, are ubiquitous in the atmosphere. So, a single layer representation is rather a crude approximation. The algorithm by Xu et al. assumes a continuous profile (and so are some others in the theoretic work). within 1M collocated pairs, will ALH comparison be different or be the same regardless one single layer or multiple layer aerosols? Can CALIPSO be helpful to identify or illustrate some cases of multiple layer of aerosols?

b. The ALH retrieval co-vary with AOD and UVAI. Xu et al. (2020) show some analysis on that. It might be interesting to plot ALH vs. UVAI for different AOD value ranges (as

Xu et al.) and see if the finding is consistent with Xu's finding.

c. It remains puzzling while 1M collocation pairs only give less 800 data points in the Figure 7. Why? Can we show all the collocated data points and find out how many percentage of TROPOMI ALH lies in +/- one sigma of the CALIOP ALH? This information is needed, and will the similar as MODIS AOD validation (which says, 76% or more data points are in +/- STD and have uncertainty of 0.05+/- 0.10AOD over land). Can such uncertainty envelope be derived?

d. The illustration case is almost exclusively for Saharan dust layer over ocean. A suggestion is to change the title of this paper to say validation or evaluation over the ocean? For this reviewer, it is bit difficulty to comprehend how well TROPOMI ALH over land, unless one case can be demonstrated.
* * *

---

## Author Comment (AC1) · 30 Jan 2020

**Reviewer comment (general)**: The manuscript entitled by "Validating TROPOMI aerosol layer height retrievals with CALIOP data" shows the initial validation results of aerosol vertical structure information from TROPOMI sensor. The aerosol vertical information is important result for the trace gas retrieval and air quality information relating to the PM2.5 etc. For this reason, the validation of aerosol layer height retrieval result is essential to publish. However, several supplements are required before the publication.

**Reviewer comment (Specific comment 1):** Although the aerosol layer information by the environment satellite mission is limited, several previous studies were investigated

including sensitivity results and methodology. Therefore, please add the reference for the aerosol height retrieval algorithm relating to next generation of environmental satellites (such as GEMS, TEMPO etc.). e.g.)

1. Choi, Wonei, et al. "Effects of spatiotemporal $O_4$ column densities and temperature-dependent $O_4$ absorption cross-section on an aerosol effective height retrieval algorithm using the O4 air mass factor from the ozone monitoring instrument." Remote Sensing of Environment 229 (2019): 223-233.

2. Kim, Mijin, et al. "Optimal Estimation-Based Algorithm to Retrieve Aerosol Optical Properties for GEMS Measurements over Asia." Remote Sensing 10.2 (2018): 162.

3. Park, Sang Seo, et al. "Utilization of O 4 slant column density to derive aerosol layer height from a space-borne UV–visible hyperspectral sensor: sensitivity and case study." Atmospheric Chemistry and Physics 16.4 (2016): 1987-2006.

4. Zoogman, P., et al. "Tropospheric emissions: Monitoring of pollution (TEMPO)." Journal of Quantitative Spectroscopy and Radiative Transfer 186 (2017): 17-39.

5. Vasilkov, A., J. Joiner, and R. Spurr. "Note on rotationalRaman scattering in the O 2 A-and B-bands." Atmospheric Measurement Techniques 6.4 (2013): 981-990.

6. Wagner, T., et al. "A sensitivity analysis of Ring effect to aerosol properties and comparison to satellite observations." Atmospheric Measurement Techniques 3.6 (2010): 1723-1751.

**Author's response:** The recommended citations have been added to the manuscript at specific sections pertaining to their relevance.

**Changes to the manuscript:** The following paragraphs include the citations requested.

Section 1:

Some notable mentions of missions that retrieve aerosol layer height are Multiangle Imaging Spectroradiometer (MISR) on board the NASA Terra satellite (Nelson et al., 2013), the Deep Space Climate Observatory (DSCOVR) mission with its Earth Polychromatic Imaging Camera (EPIC) (Xu et al., 2017, 2019), the upcoming Multi-Angle Imager for Aerosols (MAIA) mission (Davis et al., 2017), the Ozone Monitoring Instrument (OMI)on board the NASA Aura mission (Chimot et al., 2017, 2018; Choi et al., 2019), and finally the TROPOMI instrument onboard the Sentinel-5 Precursor mission (Veefkind et al., 2012). In the near future, missions like the Geostationary Environment Monitoring Spectrometer (GEMS) and the Tropospheric Emissions: Monitoring Pollution mission (TEMPO) are expected to provide aerosol height retrievals as well (Kim et al., 2018; Park et al., 2016; Zoogman et al., 2017). These instruments are examples of missions demonstrably (some theoretically, others practically) capable of retrieving aerosol layer height accurately.

Section 2.1:

The RTM in this case is a neural network model that has learned parts of a full physics RTM derived from de Haan et al. (1987), described in Nanda et al. (2019) (Section 3), which is three orders of magnitude faster than DISAMAR. In short, the atmosphere is simplified by DISAMAR in20order to reduce computational burden, and the neural network forward model is implemented for a further performance boost in an operational environment; for instance, DISAMAR ignores rotational raman scattering even though literature has shown that the oxygen A-band ring effects are sensitive to aerosol layer height (Vasilkov et al., 2013; Wagner et al., 2010). These decisions have been made in order to speed up line-by-line calculations of DISAMAR, which are the basis of the training data for its neural network counterpart. This decision is motivated by preliminary sensitivity analyses conducted by Sanders and de Haan (2016) which conclude that the effect of ignoring RRS is not significant enough to venture in its implementation into the forward model

**Reviewer comment (specific comment 2):** In page 4, Lines 34 : For the forward model simulation, the aerosol optical and physical properties based on the Henyey-Greenstein scattering phase function is insufficient. Also, the fixed single scattering albedo affects the estimation errors due to the variability of aerosol optical properties. The atmospheric layer is also too simple as we compared to the previous researches of aerosol height estimation studies. Author has to be explained the reason of simple assumption for aerosol optical and physical properties in the TROPOMI algorithm. Especially, retrieval error of aerosol height relating to the single scattering albedo and size information were reported in several previous studies.

**Author's response:** Accepted. The manuscript will add explanation for choice of aerosol model and physical parameterisations.

**Changes to the manuscript:** The following paragraph explains the various choices in aerosol properties and profile parameterisations.

The forward model parameterises aerosols with a Henyey-Greenstein scattering phase function (Henyey and Greenstein,1941) with an asymmetry factor of 0.7, a single scattering albedo of 0.95, and a fixed aerosol optical thickness for an aerosol layer parameterised by a single atmospheric layer with a 50 hPa thickness. These assumptions have to be made since very little a priori information about aerosols in a scene is known. While more complex scattering models exist, the Henyey-Greenstein model has been used for retrieving ALH when the forward model was of line-by-line nature as the number of calculations it requires is far less than a scattering model such as the Mie model. Sensitivity analyses have shown that this assumption has few ramifications (Sanders et al., 2015). Fixing the single scattering albedo is a much bigger concern; while retrievals over the ocean do not suffer for a priori errors in the single scattering albedo, retrievals over land do have large errors and non-convergences which reduce as the the viewing zenith angle increases (Nanda et al., 2018a). The choice of using 0.95 as a fixed value arises from average values derived by Dubovik et al. (2002) from long-term observations using the aerosol robotic network (AERONET). The algorithm

assumes a single aerosol layer for the entire atmosphere, within which aerosols are uniformly distributed and the aerosol volume extinction coefficient is constant. This is an important simplification to note when comparing with CALIOP profiles, since these lidar profiles have the capability to detect multiple aerosol layers. The simplicity in the aerosol profile parameterisation arises from the fact that it is impossible to know, without prior information, whether the scene consists of a single or multiple aerosol layers. While fitting of the aerosol layer pressure thickness along with the aerosol layer mid pressure does not result in large errors in the retrieved aerosol layer height, the precision of the retrieved aerosol layer mid pressure significantly deteriorates with increasing errors in the surface albedo (Sanders et al., 2015). More research has to be done before more information on the aerosol profile is retrieved from the oxygen A-band alone.

**Reviewer comment (specific comment 3):** In page 6, line 11: For the validation of ALH, author used both level 1 and 2 data of CALIPSO. If both data exists, which of the two data do you use first?

**Author's response:** We use both data at the same time. The manuscript will mention this at this specific line.

**Changes to the manuscript:** The manuscript now reads:

The data from the CALIOP instrument relevant for validating TROPOMI ALH are the level-1 backscatter profiles and the level-2 aerosol extinction profiles, which are used at the same time.

**Reviewer comment (specific comment 4):** ) In page 7: The CALIOP data has potential error to classify cloud and aerosol. For the validation, additional consideration for cloud contamination in the aerosol products of CALIOP is also important

**Author's response:** This validation study assumes that the CALIOP extinction profiles are free from cloud contamination. This assumption is incorrect, which is why additional validation study has been done with the calculated extinction heights plotted over CALIOP backscatter profile curtain plots and analysing with the eye. The combination of the two alleviates many cases where the CALIOP aerosol product might be cloud contaminated. Finally, choosing relatively cloud-free scenes also helps in ensuring that cloud contamination is not a large concern. This is clarified in the manuscript.

**Changes to the manuscript:** The manuscript clarifies the reviewer's concerns with the following changes:

... the colocation technique used in this paper. The CALIOP aerosol product might be cloud contaminated as well, but this is difficult to ascertain. Plotting $ALH_{ext}$ over curtain plots of level-1 total backscatter profiles can be used to visually discern possibly cloud-contaminated CALIOP level-2 aerosol product.

**Reviewer comment (specific comment 5):** Figure 2: From the Fishman et al. (2012) in BAMS, the reference value of aerosol layer height error is 1 km. However, only 50% of the data satisfies the error within 1 km, and the standard deviation is always larger than 1 km as author wrote in the manuscript. Compared to the expected error (1 km), the error is relatively large. Given these results, do you think the accuracy of these results is sufficient?

**Author's response:** The errors shown in this manuscript pertain to comparison of aerosol layer heights obtained by two different instruments with two separate principles. Our understanding of numbers such as expected errors pertain to errors with respect to the true aerosol layer height. This is impossible to know, as neither CALIOP nor TROPOMI retrieve the true aerosol layer height or aerosol profile, but simply express what is observed by the instrument and what is retrieved with the available information. The authors of this paper are unaware as to how these reference values by Fishman

et al. (2012) were calculated. It is also unclear what this number means, especially considering how retrieval of aerosol layer height from spectral measurements of the top of atmosphere radiance in the oxygen A-band depends on whether the aerosol plume is over a dark or a bright surface. Considering these concerns, the authors make no comment on how the accuracy of these results map to the reference aerosol layer height error values mentioned in Fishman et al. (2012).

**Changes to the manuscript:** No changes are made in the manuscript pertaining to this reviewer comment.

**Reviewer comment (specific comment 6):** Figure 7: Compared to the slope value, the Y-intercept is too large. Please discuss the reason of large positive bias of Y-intercept.

**Author's response:** Accepted.

**Changes to the manuscript:** The following clarifies the comment:

What is immediately apparent is that, while there seems to be an agreement between the two heights (indicated by the pearson correlation coefficient of 0.64, the slope of fit of 1.0 and an intercept of 0.53 km), CALIOP $ALH_{ext}$ are systematically higher than TROPOMI ALH (indicated by a y-intercept of the fit at 0.53 km). The CALIOP $ALH_{ext}$ is also higher than TROPOMI ALH almost consistently in most cases. This could possibly be due to CALIOP possibly underestimating the aerosol layer thickness due to strong attenuation of the lidar signal at the top of the aerosol layer (Rajapakshe et al., 2017), whereas TROPOMI ALH product does not suffer from such attenuation.

**Reviewer comment (specific comment 6):** In Page 11: For the further study, the author discusses to update the LER product. However, updating aerosol properties

are most important point in this study. Please add the author's opinion.

**Author's response:** Accepted.

**Changes to the manuscript:** The following amendment is made to the manuscript:

Currently, the GOME-2 surface LER product derived from Tilstra et al. (2017) is used operationally, and will eventually need to be updated with a higher resolution version possibly derived from TROPOMI itself. To that extent, owing to the boost in the computational speed of the radiative transfer calculations, the algorithm can now incorporate more complex aerosol property and profile parameterizations. Such a step will benefit the TROPOMI aerosol layer height retrieval accuracy significantly.

**Reviewer comment (technical comment 1):** In page 9 (Line 17) : correct the typo-error (4thrd -> 4th)

**Author's response:** Accepted

**Changes to the manuscript:** The typographical error has been fixed in the manuscript.

---

## Author Comment (AC2) · 30 Jan 2020

**Reviewer comment (general):** The paper presents the first attempt to validate the TROPOMI ALH data product and therefore, bears lots of research interest from the community. The work overall is sound. However, given the unprecedented observation from TROPOMI and its potential for the wide use of ALH, the paper should be revised to provide an uncertainty estimate to the community. In fact, from what is presented in the paper, the ALH at the pixel level appears to have large uncertainties. Perhaps changing 'validating' to 'first evaluating" is more appropriate for this paper – this is a suggestion though. Major recommendations are provided below.

**Author's response:** That ALH at pixel level appears to have large uncertainties is

[Figure]

a generalised statement, especially since the comparisons are against co-located CALIOP aerosol extinction heights and backscatter profiles and not the true aerosol layer height. The authors acknowledge the suggestion of renaming the paper, but see no difference between 'validating' and 'first evaluating' - albeit for a different instrument, the ALH product developed at the KNMI has been evaluated with different datasets previously. The data product is also being evaluated by a separate paper that compares it to MISR (see https://www.atmos-meas-tech-discuss.net/amt-2019-411/ for more information), which is now in the response phase. These conditions make renaming it to a first evaluation inapplicable. Alternatively, considering that the paper is a comparison of TROPOMI ALH to CALIOP aerosol heights derived from extinction profiles, the paper is renamed to — A first comparison of TROPOMI aerosol layer height to CALIOP data.

**Reviewer comment (specific 1):** While the introduction part mentioned several papers regarding ALH retrieval, it didn't go in depth to the method themselves. A few notes are highlighted below; more details can be found in Xu et al. (2017).

1. MISR offers stereo height information; this is simply done by geometric optics, and providing top height of the aerosol layer

2. Xu et al., 2017: used primarily O2 A band (I think) for ocean, while Xu et al. 2019 used O2 B band primarily over land. In both cases, these two papers demonstrate for the first time in the literature that diurnal variation of ALH can be retrieved. They are also the first to define the method to evaluate ALH from such retrievals. Somewhere, it is worthy mentioning these, for example, in the method part for equation 2, and in the analysis and discussion part regarding surface reflectance.

3. MAIA by Davis et al. This is really a theoretical work as MAIA is not launched yet. This work should be separated from the work that uses the real data, and should be lumped with other theoretical work (such as Ding et al., 2017).

Reference: Xu et al., (2018), Passive remote sensing of aerosol height, in Remote Sensing of Aerosols, Clouds, and Precipitation, edited by T. Islam, Y. Hu, A. Kokhanovsky, and J. Wang, pp.1-22, Elsevier, Cambridge, MA.

Ding et al. (2016), Polarimetric remote sensing in O2 A and B bands: Sensitivity study and information content analysis for vertical profile of aerosols, AMT

**Author's response:** Accepted. The authors have decided to mention the details that the referee suggests in the introduction section.

**Changes to the manuscript:** Changes to introduction section.

Some notablementions of missions that retrieve aerosol layer height are Multiangle Imaging Spectroradiometer (MISR) on board the NASA Terra satellite (Nelson et al., 2013) which measures aerosol height using geometric optics, the Deep Space Climate Observatory (DSCOVR) mission with its Earth Polychromatic Imaging Camera (EPIC) (Xu et al., 2017, 2019), the Ozone Monitoring Instrument (OMI) on board the NASA Aura mission (Chimot et al., 2017, 2018; Choi et al., 2019), and finally the TROPOMI instrument on board the Sentinel-5 Precursor mission (Veefkind et al., 2012). In the near future, missions like the upcoming Multi-Angle Imager for Aerosols (MAIA) mission (Davis et al., 2017), the Geostationary Environment Monitoring Spectrometer (GEMS) and the Tropospheric Emissions: Monitoring Pollution mission (TEMPO) are expected to provide aerosol height retrievals as well (Kim et al., 2018; Park et al., 2016; Zoogman et al., 2017). These instruments are examples of missions demonstrably (some theoretically, others practically) capable of retrieving aerosol layer height.

**Reviewer comment (specific 2):** The analysis part is really short in this paper. A few questions are suggested here with a hope to improve the analysis and add more 'meat' to the paper.

1. It is worth mentioning that aerosols, unlike cloud droplets, are ubiquitous in the atmosphere. So, a single layer representation is rather a crude approximation. The algorithm by Xu et al. assumes a continuous profile (and so are some others in the theoretic work). within 1M collocated pairs, will ALH comparison be different or be the same regardless one single layer or multiple layer aerosols? Can CALIPSO be helpful to identify or illustrate some cases of multiple layer of aerosols?

2. The ALH retrieval co-vary with AOD and UVAI. Xu et al. (2020) show some analysis on that. It might be interesting to plot ALH vs. UVAI for different AOD value ranges (as Xu et al.) and see if the finding is consistent with Xu's finding.

3. It remains puzzling while 1M collocation pairs only give less 800 data points in the Figure 7. Why? Can we show all the collocated data points and find out how many percentage of TROPOMI ALH lies in +/- one sigma of the CALIOP ALH? This information is needed, and will the similar as MODIS AOD validation (which says, 76% or more data points are in +/- STD and have uncertainty of 0.05+/-0.10 AOD over land). Can such uncertainty envelope be derived?

4. The illustration case is almost exclusively for Saharan dust layer over ocean. A suggestion is to change the title of this paper to say validation or evaluation over the ocean? For this reviewer, it is bit difficulty to comprehend how well TROPOMI ALH over land, unless one case can be demonstrated.

**Author's response:** The reviewer's questions are valid and definitely add to the paper. However, some of the questions posed by the reviewer are either answered in a separate paper or are beyond the scope of this current analysis. The authors respond as follows.

1. The continuous aerosol profile framework is a feature yet to be added to our radiative transfer code. While we do observe some sensitivity to the thickness

of the aerosol layer, so far we have not been able to successfully retrieve it. CALIPSO on the other hand does observe multiple layers and, while not explicitly mentioned in this paper, the comparison does show several cases with multiple aerosol layers present (especially in the selected cases).

2. This analysis is presented in a separate paper (currently under review as well) that focuses on the TROPOMI UVAI data, which is why it is not mentioned in this paper. The results from this paper are mentioned in the amended manuscript.

3. Figure 7 is an aggregate of co-locations presented in Figure 6, where the focus is on four selected cases which are visually screened for clouds. This is why there are significantly less number of data points in Figure 7. With regards to uncertainty values presented by the reviewer, Figure 7 provides numbers that represent the differences between TROPOMI and CALIOP. This is further clarified in the text.

4. Considering that the paper does present analyses of retrievals over land, it would be incorrect to further rename the paper. Citation for the comparison of TROPOMI ALH retrievals over land with other retrieval techniques is provided in the amended manuscript.

**Changes to the manuscript:** The following statements are added to the manuscript (for each of the reviewer's comments).

1. The algorithm assumes a single aerosol layer for the entire atmosphere, within which aerosols are uniformly distributed and the aerosol volume extinction coefficient is constant. This is an important simplification to note when comparing with CALIOP profiles, since these lidar profiles have the capability to detect multiple aerosol layers. The simplicity in the aerosol profile parameterisation arises from the fact that it is impossible to know, without prior information, whether the scene

consists of a single or multiple aerosol layers. While fitting of the aerosol layer pressure thickness along with the aerosol layer mid pressure does not result in large errors in the retrieved aerosol layer height, the precision of the retrieved aerosol layer mid pressure significantly deteriorates with increasing errors in the surface albedo (Sanders et al., 2015). More research has to be done before more information on the aerosol profile is retrieved from the oxygen A-band alone.

2. This altitude dependence increases with aerosol absorptions (i.e. SSA) and aerosol loading (i.e. AOD), whereas it becomes weaker over brighter surfaces where the importance of molecular scattering reduces significantly (Figure 9b). On the other hand, little altitude dependence is found for non-absorbing aerosols (i.e. SSA = 0.99). The conclusions from this synthetic experiment are replicate with real TROPOMI data in a separate manuscript, where for retrieved ALH for pixels with a UVAI greater than 1 for measurements from TROPOMI showed an increase in the correlation between ALH and UVAI for an increase in MODIS aerosol optical depth values for the same scenes. This manuscript is currently submitted to Atmospheric Chemistry and Physics and awaits review.

3. A direct comparison of the CALIOP $ALH_{ext}$ and TROPOMI ALH for these four selected cases are presented in Figure 7. For this comparison, every cloud-filtered and sun-glint-filtered TROPOMI pixel with ALH information colocated to a specific CALIOP level-2 aerosol extinction profile in Figure 6 is averaged and a standard deviation is also computed. These averaged TROPOMI ALH are then compared to the CALIOP $ALH_{ext}$, and show that TROPOMI ALH differ from CALIOP $ALH_{ext}$ by 0.53 km, with a pearson correlation coefficient of 0.64 and a slope of 1.0; CALIOP $ALH_{ext}$ are systematically higher than TROPOMI ALH (indicated by a y-intercept of the fit at 0.53 km). The CALIOP $ALH_{ext}$ is also higher than TROPOMI ALH almost consistently in most cases. This could possibly be due to CALIOP possibly underestimating the aerosol layer thickness due to strong attenuation of the lidar signal at the top of the aerosol layer (Rajapakshe et al., 2017), whereas
TROPOMI ALH product does not suffer from such attenuation.

4. A significant majority of successful the retrievals in these selected scenes are over a dark surface, owing to the bright surface albedo of the Saharan desert. The reader is point to Griffin et al. (2019) for comparison of the TROPOMI ALH retrievals over land for biomass burning aerosol plumes with the same from several other instruments including CALIOP.
* * *

---

## Editor Decision (ED1)

Technical comments for a manuscript titled "A first comparison of TROPOMI aerosol layer height to CALIOP Data" by Nanda et al.

P2

2: UV absorbing index (UVAI) → should this be 'UV aerosol index (UVAI)'?
    This definition is mixed up throughout the texts. Please see below.
32: Should include Sentinel 4 in addition to
32: for GEMS products including aerosol layer height, there is a updated reference for your consideration, for balance with Zoogman et al. of TEMPO :

Kim, Jhoon et al. (2020), New Era of Air Quality Monitoring from Space: Geostationary Environment Monitoring Spectrometer (GEMS), *BAMS*, 101, 1, doi:10.1175/BAMS-D-18-0013.1.

34: there ar → there are

P4

2: ALH - acronym not defined in main body (defined in abstract only)
7: aerosol layer height → ALH
10: Section 2) → Section 2
12: 3 → Section 3
14: section 4 → Section 4
15: 1E4-1E7 → $1x10^4$ ~ $1x10^7$
20: DISAMAR - acronym not defined. Also need a reference

P5

13: Mie model – need a reference as authors did for Henyey and Greenstein (1941)
18: AERONET – need ref. with acronym definition
23, 24: mid pressure – it was referred as 'centroid pressure' in p3:26, if my understanding is correct. Need consistency in wording

P6

6. : UV Absorbing Index – is this different from UVAI, which is UV Aerosol Index in p4:27 ? This is confusing with the definition in p2:2. If not, please use 'UVAI' as defined earlier.
16: 1e-7 → $1x10^{-7}$
19: bitwise-and – do you need '-' here?
26: receive channel → receiver channels
28: aerosol layer heights → ALHs

P7

Table 1 caption: define IODD.

    Solar zenith > 75 deg → Solar zenith angle > 75 deg

    Acronyms used in the Table should be defined: e.g. DEM, STD ..

5: lidar → LIDAR throughout the manuscript

P8

10: that aren't cloud filtered → how about 'regardless of cloud filtering, '

14~18: → This sentence is too long to read and understand. Please consider to split into two sentences, one for land and the other for ocean.

31: AOT not defined

P9

13: differing → different or difference

22: 'UVAI' was defined earlier (but need to correct the confusion mentioned earlier)

24: height of aerosol layer → ALH

27: successful the retrievals → successful retrievals

31: species → particles? components?

P10

5: aerosol layer height → ALH

10: AOT was used earlier in p8. Should be defined where it was first used.

15: inspection of figures in Figure 5 → inspection of Figure 5

20: In case such as case c, → In case c,

26: aerosol layer height → ALH

P11

1: 21.50 deg → 21.5 deg

11~12: too many 'that' … expression which result in poor readability. Very confusing. Or, at least, how about the following sentence ?

Parts of the CALIOP curtain plots for cases a, b and c suggest the existence of a possible second layer beneath the layer that is visually obvious, or that the desert dust layer extends deeper to the surface and the CALIOP signal is simply too attenuated to detect it.

31~32: on average by approximately -1 km and -0.7 km median → meant 'by approximately – 1km on average and -0.7 km as median'?

P12

1: aerosol layer height → ALH

10: can to be → can be

17: seem to not be → do not seem to be
23: aerosol layer height → ALH
24: are a very good source → is a very good source

P13
1: scipy.spatial.KDtree module → need reference
5: co-locations → need consistency in manuscript, either 'colocations' or 'co-locations'
16: 'SSA' is proportional to scattering, not absorption. 'Co-albedo' is more appropriate (Co-albedo = 1-SSA)
16: AOD → AOT has been used throughout the manuscript. Need consistency.
21: aerosol optical depth → AOD with acronym definition, but need consistency between AOT and AOD.

---

## Author Response (AR2)

Author's response to reviewer comments (report 2, second round) for the manuscript "*A first comparison of TROPOMI aerosol layer height to CALIOP data*" (amt-2019-348).

**Reviewer comment (Specific comment 1):**

1. Figure 2. the red and yellow colors should be explained in legend. blue stands for ocean, red for lad, what is the yellow-red color? need to explain this on the legends (what currently have two colors only).

2. Figure 7. what are the meaning of these colors? They are explained in the text somewhere. But it would be nice to add them in the figure caption that should be self described.

3. the abstract, as it written, is very general. it might be good to add some quantitative results you find in the abstract (in terms of mean bias and number of cases/points and time period studied).

**Author's response:**

1. There are no more than red and blue. The yellow-red part of the histogram is the non-overlapping part.

2. agreed.

3. agreed.

**Changes to the manuscript:** The following changes were added to the manuscript:

1. no changes.

2. The colors represent the cases — blue for case a, yellow for case b, green for case c and red for case d.

3. A case-by-case analysis of the data from the four selected cases (mostly around the Saharan region with approximately 800 colocated TROPOMI pixels and CALIOP profiles in Jun and December of 2018) shows that aerosol layer heights retrieved from TROPOMI using the operational Sentinel-5 Precursor Level-2 ALH algorithm is lower than CALIOP aerosol extinction heights by approximately 0.5 km. Looking at data beyond these cases, it is clear that there is a significant difference when it comes to retrievals over land, where these differences can easily go over 1 km on average.

**Editor's comments:**

P2

2: UV absorbing index (UVAI) : should this be 'UV aerosol index (UVAI)'? This definition is mixed up throughout the texts. Please see below.

**This has now been fixed.** 32: Should include Sentinel 4 in addition to 32: for GEMS products including aerosol layer height, there is a updated reference for your consideration, for balance with Zoogman et al. of TEMPO :

Kim, Jhoon et al. (2020), New Era of Air Quality Monitoring from Space: Geostationary Environment Monitoring Spectrometer (GEMS), BAMS, 101, 1, doi:10.1175/BAMS-D-18-0013.1.

**The citations have been added.**

34: there ar : there are

**this is now fixed.**

P4 2: ALH - acronym not defined in main body (defined in abstract only)

**the acronym has been added in the main body, and used throughout**

7: aerosol layer height : ALH 10: Section 2) : Section 2 12: 3 : Section 3 14: section 4 : Section 4

**fixed.**

15: 1E4-1E7 : 1x104  1x107

**fixed**

20: DISAMAR - acronym not defined. Also need a reference

**this is now added.**

P5 13: Mie model – need a reference as authors did for Henyey and Greenstein (1941)

**added.**

18: AERONET – need ref. with acronym definition

**added.**

23, 24: mid pressure – it was referred as 'centroid pressure' in p3:26, if my understanding is correct. Need consistency in wording

**centroid has been removed. it is mid pressure**

P6 6. : UV Absorbing Index – is this different from UVAI, which is UV Aerosol Index in p4:27 ? This is confusing with the definition in p2:2. If not, please use 'UVAI' as defined earlier.

**this is now changed.**

16: 1e-7 : 1x10-7

**changed.**

19: bitwise-and – do you need '-' here?

**fixed.**

26: receive channel : receiver channels

**fixed.**

28: aerosol layer heights : ALHs

**this is now changed.**

P7 Table 1 caption: define IODD.

**done.**

Solar zenith ¿ 75 deg : Solar zenith angle ¿ 75 deg Acronyms used in the Table should be defined: e.g. DEM, STD ..

**done.**

5: lidar : LIDAR throughout the manuscript

**done.**

P8 10: that aren't cloud filtered : how about 'regardless of cloud filtering, '

**done.**

14 18: : This sentence is too long to read and understand. Please consider to split into two sentences, one for land and the other for ocean.

**the sentence is shorter.**

31: AOT not defined

**done and redundant ones have been replaced with acronym.**

P9 13: differing : different or difference

**it is fixed.**

22: 'UVAI' was defined earlier (but need to correct the confusion mentioned earlier)

**fixed.**

24: height of aerosol layer : ALH

**fixed.**

27: successful the retrievals : successful retrievals

**fixed.**

31: species : particles? components?

**done.**

P10 5: aerosol layer height : ALH

**done**

10: AOT was used earlier in p8. Should be defined where it was first used.

**done.**

15: inspection of figures in Figure 5 : inspection of Figure 5

**fixed.**

20: In case such as case c, : In case c,

**done.**

26: aerosol layer height : ALH

**done.**

P11 1: 21.50 deg : 21.5 deg

**done**

11 12: too many 'that' ... expression which result in poor readability. Very confusing. Or, at least, how about the following sentence ?

Parts of the CALIOP curtain plots for cases a, b and c suggest the existence of a possible second layer beneath the layer that is visually obvious, or that the desert dust layer extends deeper to the surface and the CALIOP signal is simply too attenuated to detect it.

**this has been adopted.**

31 32: on average by approximately -1 km and -0.7 km median : meant 'by approximately – 1km on average and -0.7 km as median'?

**yes. this is fixed as suggested.**

P12 1: aerosol layer height : ALH

**done.**

10: can to be : can be

**done.**

17: seem to not be : do not seem to be

**done.**

23: aerosol layer height : ALH

**done.**

24: are a very good source : is a very good source

**The sentence is changed to:** Finally, space based LIDAR (such as the CALIOP instrument on board the CALIPSO mission) is a very good tool to retrieve aerosol vertical information to validate the TROPOMI ALH product.

P13 1: scipy.spatial.KDtree module : need reference

**a reference has been added**

5: co-locations : need consistency in manuscript, either 'colocations' or 'co-locations'

**fixed.**

16: 'SSA' is proportional to scattering, not absorption. 'Co-albedo' is more appropriate (Co-albedo = 1-SSA)

**changed to:** This altitude dependence increases with aerosol single scattering albedo)...

16: AOD : AOT has been used throughout the manuscript. Need consistency.

**fixed.**

21: aerosol optical depth : AOD with acronym definition, but need consistency between AOT and AOD.

**fixed.**

**A first comparison of TROPOMI  ALH to CALIOP data**

Swadhin Nanda[2], Martin de Graaf[1], J. Pepijn Veefkind[1, 2], Maarten Sneep[1], Mark ter Linden[1, 3], Jiyunting Sun[1, 2], and Pieternel F. Levelt[1, 2]

[1]Royal Netherlands Meteorological Institute (KNMI), Utrechtseweg 297, 3731 GA De Bilt, The Netherlands
[2]Delft university of Technology (TU Delft), Mekelweg 2, 2628 CD Delft, The Netherlands
[3]S&T Corp, Delft, The Netherlands

*Correspondence to:* Swadhin Nanda (s.nanda@tudelft.nl)

**Abstract.** The Tropospheric Monitoring Instrument's (TROPOMI) level-2 aerosol layer height (ALH) product has now been released to the general public. This product is retrieved using TROPOMI's measurements of the oxygen A-band, radiative transfer model (RTM) calculations augmented by neural networks and an iterative optimal estimation technique. The TROPOMI ALH product will deliver  ALH estimates over cloud-free scenes over the ocean and land that contain aerosols above a certain threshold of the measured UV  aerosol index (UVAI) in the ultraviolet region. This paper provides background for the ALH product and explores its quality by comparing ALH estimates to similar quantities derived from spaceborne  LIDARs observing the same scene. The spaceborne  LIDAR chosen for this study is the Cloud-Aerosol  LIDAR with Orthogonal Polarisation (CALIOP) on board the Cloud-Aerosol  LIDAR and Infrared Pathfinder Satellite Observations (CALIPSO) mission, which flies in formation with NASA's A-train constellation since 2006 and is a proven source of data for studying ALHs. The influence of the surface and clouds are discussed and the aspects of the TROPOMI ALH algorithm that will require future development efforts are highlighted.

A case-by-case analysis of the data from the four selected cases (mostly around the Saharan region with approximately 800 colocated TROPOMI pixels and CALIOP profiles in Jun and December of 2018) shows that ALHs retrieved from TROPOMI using the operational Sentinel-5 Precursor Level-2 ALH algorithm is lower than CALIOP aerosol extinction heights by approximately 0.5 km. Looking at data beyond these cases, it is clear that there is a significant difference when it comes to retrievals over land, where these differences can easily go over 1 km on average.

[revised manuscript text omitted]

de Rooij, W. A. and van der Stap, C. C. A. H.: Expansion of Mie scattering matrices in generalized spherical functions, Astronomy and Astrophysics, 131, 237–248, http://adsabs.harvard.edu/abs/1984A%26A...131..237D, 1984.

Dubovik, O., Holben, B., Eck, T. F., Smirnov, A., Kaufman, Y. J., King, M. D., Tanré, D., and Slutsker, I.: Variability of Absorption and Optical Properties of Key Aerosol Types Observed in Worldwide Locations, Journal of the Atmospheric Sciences, 59, 590–608, https://doi.org/10.1175/1520-0469(2002)059<0590:VOAAOP>2.0.CO;2, https://journals.ametsoc.org/doi/full/10.1175/1520-0469%282002%29059%3C0590%3AVOAAOP%3E2.0.CO%3B2, 2002.

Griffin, D., Sioris, C., Chen, J., Dickson, N., Kovachik, A., Graaf, M. d., Nanda, S., Veefkind, P., Dammers, E., McLinden, C. A., Makar, P., and Akingunola, A.: The 2018 fire season in North America as seen by TROPOMI: aerosol layer height validation and evaluation of model-derived plume heights, Atmospheric Measurement Techniques Discussions, pp. 1–30, https://doi.org/https://doi.org/10.5194/amt-2019-411, https://www.atmos-meas-tech-discuss.net/amt-2019-411/#discussion, 2019.

Henyey, L. C. and Greenstein, J. L.: Diffuse radiation in the Galaxy, The Astrophysical Journal, 93, 70, https://doi.org/10.1086/144246, http://adsabs.harvard.edu/doi/10.1086/144246, 1941.

Herman, J. R., Bhartia, P. K., Torres, O., Hsu, C., Seftor, C., and Celarier, E.: Global distribution of UV-absorbing aerosols from Nimbus 7/TOMS data, Journal of Geophysical Research: Atmospheres, 102, 16 911–16 922, https://doi.org/10.1029/96JD03680, http://doi.wiley.com/10.1029/96JD03680, 1997.

Holben, B. N., Eck, T. F., Slutsker, I., Tanré, D., Buis, J. P., Setzer, A., Vermote, E., Reagan, J. A., Kaufman, Y. J., Nakajima, T., Lavenu, F., Jankowiak, I., and Smirnov, A.: AERONET—A Federated Instrument Network and Data Archive for Aerosol Characterization, Remote Sensing of Environment, 66, 1–16, https://doi.org/10.1016/S0034-4257(98)00031-5, http://www.sciencedirect.com/science/article/pii/S0034425798000315, 1998.

5 Ingmann, P., Veihelmann, B., Langen, J., Lamarre, D., Stark, H., and Courrèges-Lacoste, G. B.: Requirements for the GMES Atmosphere Service and ESA's implementation concept: Sentinels-4/-5 and -5p, Remote Sensing of Environment, 120, 58–69, https://doi.org/10.1016/j.rse.2012.01.023, http://linkinghub.elsevier.com/retrieve/pii/S0034425712000673, 2012.

IPCC: Clouds and Aerosols, in: Climate Change 2013 - The Physical Science Basis, pp. 571–658, Cambridge University Press, Cambridge, https://doi.org/10.1017/CBO9781107415324.016, http://ebooks.cambridge.org/ref/id/CBO9781107415324A024, 2014.

10 Kim, J., Jeong, U., Ahn, M.-H., Kim, J. H., Park, R. J., Lee, H., Song, C. H., Choi, Y.-S., Lee, K.-H., Yoo, J.-M., Jeong, M.-J., Park, S. K., Lee, K.-M., Song, C.-K., Kim, S.-W., Kim, Y. J., Kim, S.-W., Kim, M., Go, S., Liu, X., Chance, K., Chan Miller, C., Al-Saadi, J., Veihelmann, B., Bhartia, P. K., Torres, O., Abad, G. G., Haffner, D. P., Ko, D. H., Lee, S. H., Woo, J.-H., Chong, H., Park, S. S., Nicks, D., Choi, W. J., Moon, K.-J., Cho, A., Yoon, J., Kim, S.-k., Hong, H., Lee, K., Lee, H., Lee, S., Choi, M., Veefkind, P., Levelt, P. F., Edwards, D. P., Kang, M., Eo, M., Bak, J., Baek, K., Kwon, H.-A., Yang, J., Park, J., Han, K. M., Kim, B.-R., Shin, H.-W., Choi, H.,

15 Lee, E., Chong, J., Cha, Y., Koo, J.-H., Irie, H., Hayashida, S., Kasai, Y., Kanaya, Y., Liu, C., Lin, J., Crawford, J. H., Carmichael, G. R., Newchurch, M. J., Lefer, B. L., Herman, J. R., Swap, R. J., Lau, A. K. H., Kurosu, T. P., Jaross, G., Ahlers, B., Dobber, M., McElroy, C. T., and Choi, Y.: New Era of Air Quality Monitoring from Space: Geostationary Environment Monitoring Spectrometer (GEMS), Bulletin of the American Meteorological Society, 101, E1–E22, https://doi.org/10.1175/BAMS-D-18-0013.1, https://journals.ametsoc.org/doi/full/10.1175/BAMS-D-18-0013.1, 2019.

[revised manuscript text omitted]

**Figure 9.** Sensitivity analysis of UV aerosol index to show the influence of different aerosol properties on the UVAI. The aerosols in these analyses have a Henyey-Greenstein scattering phase function with an asymmetry factor of 0.7, an angstrom exponent of 1.0, the viewing zenith angle is $0°$, the solar azimuth angle and the viewing azimuth angles are $0°$ and $60°$ respectively, the surface pressure is 1013 hPa, and for this specific case, the solar zenith angle $\theta_0$ is $30°$. The y-axis is the UVAI for 340 nm and 380 nm, whereas the x-axis is the height of the geometric centroid of the aerosol layer in hPa (Haer). The legend in each of the plots describe the different configurations chosen for these sensitivity analyses. **(a)** looks into the sensitivity of UVAI with a fixed surface albedo of 0.05, and **(b)** does the same for a fixed  AOT of 0.5.